# An Object-Centric Hierarchical Pose Estimation Method Using Semantic High-Definition Maps for General Autonomous Driving

**DOI:** 10.3390/s24165191

**Published:** 2024-08-11

**Authors:** Jeong-Won Pyo, Jun-Hyeon Choi, Tae-Yong Kuc

**Affiliations:** Department of Electrical and Computer Engineering, College of Information and Communication Engineering, Sungkyunkwan University, Suwon 16419, Republic of Korea; choijunhyeon@skku.edu (J.-H.C.); tykuc@skku.edu (T.-Y.K.)

**Keywords:** autonomous driving, object recognition, place recognition, pose estimation, high-definition map

## Abstract

To achieve Level 4 and above autonomous driving, a robust and stable autonomous driving system is essential to adapt to various environmental changes. This paper aims to perform vehicle pose estimation, a crucial element in forming autonomous driving systems, more universally and robustly. The prevalent method for vehicle pose estimation in autonomous driving systems relies on Real-Time Kinematic (RTK) sensor data, ensuring accurate location acquisition. However, due to the characteristics of RTK sensors, precise positioning is challenging or impossible in indoor spaces or areas with signal interference, leading to inaccurate pose estimation and hindering autonomous driving in such scenarios. This paper proposes a method to overcome these challenges by leveraging objects registered in a high-precision map. The proposed approach involves creating a semantic high-definition (HD) map with added objects, forming object-centric features, recognizing locations using these features, and accurately estimating the vehicle’s pose from the recognized location. This proposed method enhances the precision of vehicle pose estimation in environments where acquiring RTK sensor data is challenging, enabling more robust and stable autonomous driving. The paper demonstrates the proposed method’s effectiveness through simulation and real-world experiments, showcasing its capability for more precise pose estimation.

## 1. Introduction

Among numerous research topics, autonomous driving has recently become one of the most actively researched subjects, particularly accelerating with advancements in deep learning technologies. As a result, automobile manufacturers are increasingly focusing on developing vehicles that can drive autonomously without human intervention, moving beyond just assisting human drivers through Advanced Driver Assistance Systems (ADAS). However, developing fully autonomous vehicles remains challenging, with many unresolved issues. These issues are not just technical, but also institutional and legal, requiring significant effort and time to address. This leads companies like Waymo (Waymo LLC, Mountain View, California, USA), GM (General Motors Corporation, Detroit, Michigan, USA), and Tesla (Tesla Motors, Inc., Austin, Texas, USA) to dedicate substantial efforts to develop fully autonomous vehicles, yet they need help achieving complete autonomy.

As we know, multiple technologies have to be integrated into a single vehicle to achieve full vehicle autonomy. Among these technologies, the vehicle’s ability to perceive its current state and surroundings is the most crucial. For this purpose, various sensors such as cameras, LiDAR, radar, ultrasound, and GPS are utilized. Recently, with the advancement of deep learning, perception technologies using sensor data have also progressed rapidly, significantly enhancing performance and reliability compared to previous levels. These advancements have greatly contributed to lane detection [1,2,3,4,5,6,7,8,9,10] and object detection around the vehicle [11,12,13,14,15,16]. Thanks to these advancements in autonomous driving, higher levels of autonomous driving research are being conducted and immediately applied to commercial vehicles.

To further differentiate levels of autonomy in autonomous driving, ref. [17] classified autonomous driving into six levels. Levels 0 to 2 require human intervention, meaning the driver must be seated in the driver’s seat and be ready to take control in emergencies. Levels 3 to 5 represent higher degrees of autonomy, from conditional to complete independence. Commercial vehicles generally fall into the Level 1 to 2 range, corresponding to Advanced Driver Assistance Systems (ADAS). Many researchers conduct extensive studies to achieve Level 4 and above, representing highly to fully autonomous driving. Level 3 and above require the autonomous system to manage all driving tasks under certain conditions without driver intervention. In other words, the vehicle must be capable of stable autonomous driving without any issues when the system is active.

Autonomous driving involves numerous variables, and these variables, whether large or small, inevitably influence the performance of autonomous driving. Despite many variables, autonomous driving must be performed stably to be considered Level 4 and above. While various elements comprise the independent driving system, this paper focuses on vehicle pose estimation. Methods for estimating the vehicle’s pose are diverse, with the most common approach using Real-Time Kinematic (RTK) sensors, which allow the acquisition of the vehicle’s pose without additional computations. Unlike conventional GPS sensors, RTK sensors can provide highly accurate location information with only a few centimeters of error. However, these sensors require good signal sensitivity, and if not available, may result in errors comparable to those of regular GPS sensors or make data acquisition impossible; such situations often occur in indoor spaces, areas with obstructed ceilings, or environments with solid signal reflection. Various studies have been proposed to address the signal problem mentioned above [18,19,20,21,22], most of which aim to complement inaccuracies in RTK sensors by utilizing feature information from other sensors. However, these methods may accumulate errors if RTK sensors remain consistently inaccurate.

This paper presents a novel approach to the critical need for a robust autonomous driving system capable of Level 4 and above autonomy. To address the challenge posed by inaccurate pose estimation due to the limitations of RTK sensors in specific environments like tunnels or under bridges, we propose a unique solution using a semantic high-definition (HD) map integrated with object-centric features to enhance pose estimation accuracy where RTK data are unreliable. This method significantly improves autonomous driving stability and reliability in diverse environmental conditions, as simulations and real-world experiments demonstrate. In this paper, we composed a semantic HD map with object registration (Section 3), proposed how to make object-centric features (Section 4.1), how can we recognize place using these features (Section 4.2), presented pose estimation method with features and places (Section 4.3), and experimented in simulations and real environments we targeted (Section 5) to ensure our system works.

## 2. Related Works

Understanding the surrounding environment is an essential part of autonomous driving systems. Autonomous vehicles recognize the environment through surrounding sensors, just as humans recognize their surroundings as objects or situations immediately with their eyes. The most used method is semantic segmentation, which recently has achieved considerable improvements with deep learning. To solve the misclassification cases in network-based semantic segmentation using images, Liu et al. [11] presented a severity-aware reinforcement learning using the Wasserstein distance between the semantic segmentation label of the image extracted from the Carla simulator [23]. Chen et al. [12] presented a semantic segmentation method in urban scenes using reality-oriented adaptation networks (ROAD-Net) to convert synthetic data acquired in a virtual environment to an actual data domain. Moreover, Li et al. [13] proposed an Efficient Symmetric Network (ESNet) of a real-time semantic segmentation model for autonomous driving. In addition to images, studies on semantic segmentation using LiDAR have also been conducted. Aksoy et al. [14] developed a semantic segmentation network by constructing an encoder–decoder structured network of 3D LiDAR point clouds. In contrast, Cortinhal et al. [15] showed a more robust 3D LiDAR semantic segmentation network composed of applying the context module to the front part of the SalsaNet [15] constructed above. Moreover, researchers have also conducted sensor fusion methods to recognize the surrounding environment. Florea et al. [16] presented a real-time perception component based on the low-level fusion between point clouds and semantic scene information. Furthermore, Hemmati et al. [24] introduced an adaptive system with a hardware-software co-design on Zynq UltraScale+ MPSoC that detects pedestrians and vehicles in different lighting conditions on the road.

As described above, current environmental recognition methods, such as lane recognition and semantic segmentation, have achieved many results, according to active research in deep learning. One of the most critical factors in such deep learning networks is the collection of datasets and their labeling. However, it is challenging to construct a dataset because of time, money, and human resources constraints. Therefore, we generally teach networks to utilize public datasets matching our intended purposes. Recently, many datasets related to autonomous driving have been created and published. Caesar et al. [25] collected data with six cameras, five radars, and one LiDAR every 20 s in 1000 scenes and published a fully annotated dataset with 3D bounding boxes for 23 classes of eight attributes. Instead of the existing simple and small amount lane dataset, Pan et al. [26] published a dataset of 133,235 labeled images for lanes with various shapes extracted from multiple scenarios and performed lane detection with spatial CNN (SCNN) using a 3-D kernel tensor. Yu et al. [27] collected 100 K driving videos from various weather, scenes, and day hours, which were annotated by scene tagging, object bounding boxes, drivable areas, lane markings, and full-frame instance segmentation.

The most common way to perform place recognition is to handle images. Since the image projects everything visible in the scene, extracting features or recognizing objects is easy. Most methods for performing place recognition using images are the descriptor extraction or scene graph method. Neuber et al. [28] composed a place description cell from the minicolumn network (MCN) through an image binary vector that is represented as NetVLAD [29]. Peng et al. [30] generated an image descriptor from an attention-aware feature embedding using the semantic prior method to focus on the critical local features. Garg et al. [31] presented a local descriptor composed of local semantic tensors to aware invariant features, regardless of directions or day and night. Chancan et al. [32] perform the place recognition process through the global image descriptor associated with 2D global position using recurrent neural networks (RNNs). Ozdemir et al. [33] estimated place through echo state networks (ESN) using NetVLAD features from an image sequence. Woo et al. [34] generated a scene graph from an image using the geometric encoding and relationship embedding method. Also, Yang et al. [35] presented an attentional graph convolutional network (aGCN) with a relation proposal network (RePN) to generate a scene graph. Zellers et al. [36] solved this scene graph generation problem with bidirectional LSTM. JiZhang et al. [37] proposed a relationship detection network (RelDN) with graphical contrastive losses to cope with the confusing classification problem. Li et al. [38] suggested an end-to-end scene graph generation method with region captioning. Arandjelovi et al. [39] composed a network with a vector of locally aggregated descriptors (VLAD) to generate invariant global descriptors. Hausler et al. [40] composed a patch-level descriptor with a more robust invariant feature than the NetVLAD descriptor. Gu et al. [41] presented a scene graph generation process to refine features using external knowledge and image reconstruction. Cong et al. [42] performed scene graph generation with spatial and temporal relationship encoder. Cui et al. [43] composed a context-dependent diffusion network (CDDN) with a work semantic graph and spatial scene graph to generate relationships. Zareian et al. [44] presented a commonsense graph to generalize the scene graph from an image. Suhail et al. [45] presented an energy value to reduce reasoning errors or biases during the scene graph training.

In addition to the place recognition method through images, the place recognition method using the point cloud is also being actively studied. Since the point cloud has distance information, acquiring features of a different type from the picture is possible. Vidanapathirana et al. [46] presented a global descriptor generation using point cloud through second-order pooling and Eigen-value power normalization. Fan et al. [47] generated rotation invariant features from the point cloud using an attentive rotation invariant convolution (ARIConv) network. Uy et al. [29] combined PointNet [48] and NetVLAD [39] to perform the point cloud based place recognition. Wang et al. [49] presented end-to-end object recognition using dynamic graphs CNN (DGCNN) with k-NN feature graph. Chen et al. [50] proposed a network for finding the equivariant of point cloud features using reduced separable point convolution (SPConv) and invariant features through an attentive fusing mechanism. Komorowski et al. [51] generated a descriptor fused with lidar and an image and can be used as an application for loop closure.

Another critical element of autonomous driving is determining a vehicle’s current location. When people are driving, they can decide on their location and drive with the help of the GPS and navigation system installed in the vehicle. Similarly, autonomous vehicles determine their location using GPS sensors and HD maps instead of navigation systems. The difference between people and an autonomous vehicle is that people have no hindrance to driving, even if they only know the approximate location. In the case of an autonomous vehicle, it is only possible to drive when the location is precisely determined. However, deciding the accurate location requires a precise GPS sensor, which is very expensive. Therefore, many studies have been conducted to improve the performance by combining a general GPS sensor with a camera sensor. Chen et al. [18] developed a GNSS-Visual-ORB-SLAM (GVORB) method that combines low-cost GNSS and ORB-SLAM [19] for a monocular camera to compensate for the shortcomings of the GNSS sensor. Cai et al. [20] showed the correct localization method of the vehicle using the EKF with the distance between the vehicle location from the GPS sensor and the discrepancies between the lanes in the HD map and those detected by the monocular camera. In addition to camera sensors, many studies have been conducted to improve the accuracy of IMU sensors. Lee et al. [21] improved the localization accuracy through the EKF using the information of the driving lane and stop lane from the lane recognition [52] through a monocular camera to refine errors occurring in GPS, IMU, and onboard odometers. Chu et al. [22] presented the EKF method with a combined monocular camera, IMU, and GNSS to solve problems such as signal attenuation, reflections, or blockages from GNSS and accumulated errors over time from IMU using motion estimation information with SIFT and RANSAC.

After all of the driving functions of the autonomous driving system are composed, the autonomous driving system has to be applied to the vehicle in the actual environment without any problems. In contrast to the individual functions of the autonomous driving system, the methods for using the system in the actual vehicle also have many functions. Chen et al. [53] comprehensively analyzed end-to-end autonomous driving systems, focusing on the latest methodologies and challenges in achieving robust, safe driving. Additionally, Nawaz et al. [54] discussed enhancing the robustness of autonomous vehicles by integrating advanced sensor technologies with AI for improved navigation and obstacle avoidance in complex environments. For generalization in the actual environment, Wang et al. [55] proposed multi-modal foundation models to enhance feature representation, significantly improving generalization and robustness in diverse driving conditions. Also, Gao et al. [56] introduced a semantic masked world model to improve sample efficiency and robustness in urban driving, leveraging semantic filters to extract relevant driving features from sensor data. To perform autonomous driving more safely in the actual environment with artificial intelligence (AI), additionally, Mishra et al. [57] surveyed irregular situations in various AI domains, proposed potential solutions, and discussed safety and privacy issues in autonomous vehicles to improve the robustness and adaptability of AI systems in real-world scenarios. Moreover, they suggested a situation-aware traffic control hand-gesture recognition system for autonomous driving when irregular situations occur, and the human directly manages those situations by hand-gesture [58]. In addition, communication between the base station and autonomous vehicles is crucial for managing autonomous vehicles from outside the vehicle. Qiong et al. [59] proposed an intelligent vehicular node with an optimal minimum contention window (MCW) using deep-Q-learning (DQN) for vehicle-to-infrastructure (V2I) communications with the base station to improve age of information (AoI) fairness.

## 3. Semantic HD Map

Autonomous vehicles studied recently inherently possess high-definition (HD) maps for the regions they intend to navigate. HD maps store fundamental road information, such as lanes, traffic signals, and road signs. These details convey the necessary information for the vehicle’s autonomous operation. Additionally, HD maps can accommodate more extensive information based on specific requirements. However, given the diverse applications of HD maps, actively utilizing all the information they contain is uncommon. This paper employs selective information from HD maps, focusing on specific details relevant to the research objectives. This chapter briefly describes the information in conventional HD maps, emphasizing the subset of data utilized in this study and introducing additional information incorporated explicitly for this research.

### 3.1. Nodes and Links

In HD maps, information about lanes and the trajectories of the ideal paths when a vehicle optimally navigates the road are stored. These trajectories are primarily depicted around the center between lanes, and encompass all feasible road vehicle travel paths. In HD maps, these paths are distinguished by specific points, denoted as nodes. The criteria for node differentiation are typically associated with the vehicle’s path before and after curves, the creation of intersections, and when the distance exceeds a specific threshold. However, this differentiation is not absolute, and can vary based on the objectives. Once nodes distinguish all paths, these paths can be represented as detailed paths connecting nodes, referred to as links, as shown in Figure 1. Each link can be expressed as a set of multiple points, termed link points.

As we already know, vehicles generally travel in only one direction along the given path, unless it is a particular road like an alley. In other words, each path has a specified direction of travel, and accordingly, each node and link also reflects the direction of travel. As shown in Table 1, each node and link contains substantial information. Although the data stored in nodes and links in the HD map is more extensive than in Table 1, this paper utilizes only the necessary information. As observed in Table 1, this paper uses the unique ID and point information stored in each node and the RoadType, FromNodeID, ToNodeID, and points information stored in each link. The node’s ID and point represent the node’s unique ID and the node’s latitude–longitude coordinates, respectively. The link’s ID also denotes the link’s unique ID, and the RoadType, as shown in Table 1, includes type values for General roads, Tunnel, Bridge, Underground roads, and elevated highways. R_LinkID and L_LinkID represent the IDs of each link when there are changeable lanes on both sides of the current link. As mentioned earlier, since each link has a specified direction, they also have the FromNodeID, indicating the node where the link starts and the ToNodeID indicating the node where it ends. Finally, the points are a set of link points, and each link point has latitude–longitude values. Notably, the order in which link points are stored corresponds to the direction of travel of the link.

### 3.2. Registered Objects

Essentially, the HD map contains information necessary for vehicle navigation, including data that can be utilized as objects. Among these objects, this paper selects those that can be perceptually identified through sensors, as shown in Figure 2. In reality, safety signs, surface marks, traffic lights, and speed bumps, as chosen in this paper, encompass a much broader and more detailed range of information than presented in Table 2. However, this paper aims to simplify and efficiently utilize this information rather than delve into the specifics. For instance, despite having more than four diverse types, safety signs are categorized as a single type, and surface marks, with over ten variations, are simplified by distinguishing only arrow or crosswalk shapes. Similarly, traffic lights and speed bumps are each identified as a single type, disregarding the numerous shapes and types they may come in. This simplification aids in reducing complexity and streamlining the perception of objects when utilizing sensor data in real-world environments.

### 3.3. Additional Objects

This paper aims to develop a pose estimation method for autonomous driving at Level 4 and above for General Autonomous Driving. As previously explained in Section 3.2, for Level 4 and above autonomous driving, the vehicle must be able to estimate its position, even when it cannot acquire valid signals from RTK sensors. Segments, where RTK sensor signals are unavailable, are typically found in areas such as beneath elevated highways, tunnels, and urban centers, where receiving GPS signals is challenging. Thus, the objects we need to register newly will be objects perceivable within such segments and meet the conditions mentioned above. Since the verification is focused on a specific region in this paper, additional object registration prioritizes areas where GPS signals are challenging to obtain.

In this paper, as shown in Figure 3, we have registered a total of 10 types of objects: Directional Signs, Road Signs, Lane Control Signs, Emergency Lights, Jet Fans, Fire Extinguisher Lights, Support Columns, Reflection Lights, Street Tree, and Ground Transformer. Each object is selected based on satisfying at least one previously mentioned condition. For instance, the Directional Sign is a sign that informs of an imminent sharp curve in the forward direction, allowing for the estimation of a specific location. Road signs provide information about destinations that can be reached when proceeding in the driving direction, intermittently present in particular segments. The Lane Control Sign indicates whether the current lane is drivable, mainly found within tunnels in the target area. Emergency Light signals the location of commonly known emergency exits, visible only within tunnels while driving. Jet Fan similarly exists exclusively to circulate isolated air within tunnels. Fire Extinguisher Light indicates the location of fire extinguishers, typically visible only within tunnels during driving. Support Column refers to pillars supporting bridges, signifying the presence of the structure and repeating at specific intervals. A Reflection Light is a reflective light on the center median in dark highways, repeating at specific intervals above the highway. Street trees repeat at specific intervals within urban areas, and ground transformers represent distribution boards on the ground in urban areas. As explained, all the objects are aligned with the intended purpose.

In this paper, the registration of these objects is performed manually. Given our proposed method, the more accurate the position of the registered objects, the more precise the pose estimation becomes. The criteria for registering information in the HD map vary by country and institution. However, in this paper, we followed criteria such as Table 3 when manually registering objects. As seen in Table 3, when manually registering, the absolute accuracy of the object points in this paper should fall within a 0.1 m error in both horizontal and vertical positions within a 95% confidence interval, and we ensured that the object points did not exceed a maximum of 0.2 m.

As shown in Table 4 and Table 5, the selected objects have been registered in the semantic HD map. These registration models are referenced by the Triplet Ontological Semantic Model (TOSM) [60], but modified to fit our purpose in this paper. TOSM claims three data types for each object: Symbolic, Explicit, and Implicit, as shown in Table 4. Furthermore, between objects, they have relationships as shown in Table 5. All information inside there is re-formulated for this paper. Similar to previously registered objects in the original HD map, each object has a unique ID and a Type value corresponding to its object type, as shown in Table 6. Notably, the objects we added are registered based on the object’s center point as its coordinate, as depicted in Figure 3. This choice is justified for two reasons: Firstly, the selected objects are relatively small, so registering them based on the center point is not overly restrictive. Secondly, the criteria for extracting object information for comparison after object recognition in this paper are also based on the object’s center point.

The information on registered objects in the HD map is our most important ingredient for object-centric estimation. We recognize objects in a particular place, and then estimate our actual place on the HD map using our proposed features from the recognition results. It means that our proposed method can estimate two places that have the same features as the same place. For example, as shown in Figure 4, the same objects, like streetlights, are repeated a long way, like on a particular highway road. In this case, we may incorrectly estimate if the object recognition results and generated features are more similar to a specific place than the actual place where the vehicle is placed because they are almost identical, as shown in Figure 4. To solve this problem, we added the above three properties to prevent confusion: Group, Count, and isRepeated. isRepeated shows that the object is a repeated object on a continuous driving path, and Group presents a number of the repeated group when isRepeated is true. Furthermore, Count shows the n-th number within the group. Following this information, fromObjectID and toObjectID in Table 5 are defined, as shown in Figure 4, by the driving path directions. Then, other properties are defined similarly to the original TOSM.

## 4. Object-Centric Hierarchical Pose Estimation

Various functionalities need to be integrated to construct an autonomous driving system. Within the autonomous driving system, there are inherent and complex functionalities, as illustrated in Figure 5 from [61], including perceiving the environment, estimating the vehicle’s pose, determining the vehicle’s state, composing the vehicle’s driving path, creating detailed paths, and sending commands to drive the vehicle. Estimating the vehicle’s pose is considered the most prioritized among all these functionalities. This paper focuses on performing the vehicle’s pose estimation corresponding to the block indicated in Figure 5.

The objective of this paper is to estimate the pose of a vehicle based on objects. The pose estimated in this paper aims to be in two dimensions, primarily due to the nature of the HD map used, which is latitude and longitude-based and lacks height information. The overall structure of the proposed object-based pose estimation method in this paper is depicted in Figure 6. As explained in Section 3, the composed semantic HD map is used for pose estimation, and the process involves comparing object information recognized through sensor data to estimate the pose. In the offline phase shown in Figure 6, object-centric features are precomputed using information stored in the HD map. Since the information in the HD map does not change once registered, it is more efficient to precompute all features offline and use them when needed, rather than generating features online for comparison. The vehicle’s pose is estimated in two main steps in the online phase. The first step is for the vehicle to recognize its current location. The second step is to estimate its pose based on this location. In the first step, as mentioned earlier, objects are identified through sensor data, and object-centric features are generated based on this information. These features are then compared to all candidate object-centric features where the vehicle could be positioned on the HD map, determining the vehicle’s current location. In the second step, based on the recognized location, a more accurate pose of the vehicle is estimated. With knowledge of the current location, estimating the pose based on this location allows for a more precise pose estimation. This section details how object-centric features are constructed, object-centric recognition is performed, and how this information is used to perform object-centric pose estimation, ultimately leading to the estimation of the vehicle’s pose.

### 4.1. Object-Centric Feature

There are various methods for estimating the pose of a vehicle, but one of the most widely used approaches is the feature-based comparison method. However, there are differences in research methodologies; a common practice involves constructing features from sensor data and performing comparisons based on these features. With numerous studies conducted, various feature extraction methods have been developed, achieving significant performance [3,19,20,39,40,46]. Naturally, the accuracy of feature comparison increases with higher-quality features and a larger quantity of features. The quality of features depends on how well the features exhibit invariance properties concerning the target. Due to variations in sensor types and characteristics, different sensor data may be obtained, even from the exact location. If we can extract consistent features despite such variations, the accuracy of comparisons will be higher. These features typically constitute a vector with multiple values. As expected, higher-dimensional features provide a broader range for representing feature space, allowing for more detailed feature representations. However, as feature dimensionality and quantity increase, the computational cost for feature comparison also rises.

This paper aims to extract semantic features, a higher-level concept, rather than directly generating features from sensor data. As mentioned earlier, extracting features directly from sensor data inevitably influences sensor types and characteristics, regardless of the quality of the feature extraction method used. In this paper, the goal is to minimize the impact of sensor characteristics while constructing conceptual features. With the rapid advancement of deep learning, object recognition through sensor data has achieved remarkable success [5,11,12,13,14,15,16,24,48,49]. Furthermore, object recognition through deep learning is robust against noise and exhibits high viewpoint invariance, surpassing traditional methods. As it possesses the invariance properties mentioned earlier, this paper uses object recognition information to construct features. This section provides a detailed explanation of how the object-centric features we constructed are formulated.

#### 4.1.1. Link Point Interpolation

As mentioned earlier, the first task performed in the proposed method of this paper is place recognition. For place recognition, it is necessary to define places on the semantic HD map that can be considered locations. Definitions of places may vary between studies, but in this paper, the places are defined as individual link points. As previously mentioned, all drivable paths for a vehicle are represented as nodes and links, where each link consists of a set of consecutive points called link points, representing the path along the driving route in the HD map. Based on these link points, the visible surrounding environment also changes as they change with the link points’ variation. Even if the change in link points is minimal along the direction of the link, the surrounding environment does change. Thus, each link point can be considered a different place.

The estimation of the vehicle’s pose in this paper is structured stepwise, first finding the place and then estimating the precise location based on that place. Therefore, the more accurate the place recognition, the more accurate the location estimation. Accurate place recognition leads to more precise location estimation. However, factors other than accurate place recognition can also increase accuracy in location estimation. One such factor is constructing places with as much detail as possible. In this paper, recognizing a place means determining the link point to which the vehicle is closest. Naturally, the closer the vehicle is to a specific link point, the easier it is to estimate the location. However, since the link points within existing HD maps are often manually generated, they may need to be evenly spaced. Sometimes, despite the length, even in long straight links, there may be only one or two link points. Of course, minimizing the number of points for each link contributes to the HD map’s lightweight and faster processing speed. However, for place recognition in this paper, if the recognition is performed based on the existing link points, the closest link point to the vehicle may be considered the optimal place, even if it is several meters away.

The left side of Figure 7 shows link points in a specific region of the existing HD map, where each color represents a different link. As shown in Figure 7, the points are not evenly spaced, and there are noticeable gaps in severe cases. If place recognition is performed while driving through such long gaps between link points, even with accurate recognition, one of the disconnected link points may be represented as the result of place recognition. If the vehicle’s pose estimation is then performed in this state, the possibility of not finding the accurate position of the vehicle increases significantly. This situation is different from the desired result for place recognition. To overcome this, we performed interpolation between link points, as shown on the right side of Figure 7. As seen in Figure 7, the link points obtained through interpolation on the HD map are more finely represented than the original HD map link points. This interpolation process suggests that the intervals of the places to be estimated have also been more finely divided. As is, it claims that we can perform more accurate place recognition with more precise divisions.

#### 4.1.2. Vehicle Heading Based On Link Points

Ultimately, this paper aims to estimate the vehicle’s pose. As mentioned earlier, since the HD map we use provides information based on latitude and longitude, our objective is to estimate the position and orientation of the vehicle in a two-dimensional plane. As explained in the preceding section, we designated each link point as a distinct location. However, the link point only provides information about the position of the current location and does not contain information about the direction in which the vehicle is currently traveling. Regardless of autonomous vehicles’ sensor placement or performance, it is crucial to establish the reference direction for the current location. Comparing objects based solely on their information without a reference direction might lead to recognizing entirely different places with similar object structures.

While it is possible to set the reference direction based on latitude and longitude, this is not convenient for comparison, since it is not charged according to the vehicle’s perspective. Each link point already possesses a reference direction, as mentioned earlier, considering each link is already designed based on the ideal direction of vehicle travel. Therefore, the reference direction for each link point is as follows:(1)$$L=\{l_{1},l_{2},\ldots ,l_{n_{l}}\}$$
(2)$$l_{i}=\left\{l_{i}^{\mathrm{ID}},l_{i}^{\mathrm{RT}},l_{i}^{\mathrm{RID}},l_{i}^{\mathrm{LID}},l_{i}^{\mathrm{FNID}},l_{i}^{\mathrm{TNID}},l_{i}^{P}\right\}$$
(3)$$l_{i}^{P}=\left\{l_{i}^{p_{1}},l_{i}^{p_{2}},\ldots ,l_{i}^{p_{n_{l_{i}}}}\right\}$$
(4)$$h_{i}^{P}=\left\{h_{i}^{p_{1}},h_{i}^{p_{2}},\ldots ,h_{i}^{p_{n_{l_{i}}}}\right\}$$
(5)$$h_{i}^{p_{i}}=\frac{l_{i}^{p_{i+1}}-l_{i}^{p_{i}}}{∥l_{i}^{p_{i+1}}-l_{i}^{p_{i}}∥}$$

From Equations (Equation 1) to (Equation 3), *L* is the total link, $l_{i}$ is each link, and $l_{i}^{P}$ is the set of points in $l_{i}$. Also, each link has its information RT, RID, LID, FNID, TNID: RoadType (road type), R_LinkID (link ID of the right side), L_LinkID (link ID of the left side), FromNodeID (node ID the like is started), ToNodeID (node ID the like is ended), respectively. Here, $h_{i}^{P}$ in Equation (Equation 4) is all heading vectors according to link points, and Equation (Equation 5) presents to calculate each normalized heading vector. In this way, each link point has an associated reference direction known as a heading, as shown in Figure 8.

#### 4.1.3. Vehicle Perception Coverage

Autonomous vehicles come in various forms, depending on each vehicle’s purpose. Consequently, sensors’ types, configurations, and placements vary significantly among vehicles. The ideal autonomous vehicle would cover the maximum environmental range in all directions around the vehicle. However, this is constrained by cost and efficiency considerations, so not all autonomous vehicles adopt such a configuration. Hence, the types and placements of sensors on autonomous vehicles differ and, consequently, the sensor coverage is determined accordingly. Naturally, autonomous vehicles can only recognize objects within their sensor coverage. This paper assumes we know the sensor coverage for the autonomous vehicles targeted in advance.

The reason for pre-knowing sensor coverage in this paper is explained in the context of minimizing unnecessary operations for feature extraction, as mentioned in the overall structure explanation of Figure 6. To precompute object-centric features using the HD map stored information in the offline process, we need to know the sensor coverage of the autonomous vehicles we are targeting in advance. While the sensor coverage of a vehicle is typically represented in 3D, for consistency with the 2D projection of the HD map used in this paper, we simplify the sensor coverage by projecting it in 2D.

Figure 9 illustrates an example of a vehicle’s sensor coverage and the associated object information. The depicted sensor coverage assumes a vehicle with a 120-degree sensor coverage in front of the vehicle. Since we have previously calculated the vehicle’s heading for each link point, it is possible to set the vehicle’s ideal travel direction, as shown in Figure 9, adjusting the sensor coverage accordingly. The points in Figure 9 represent objects registered in the HD map. The sensor coverage based on the heading from each link point can be expressed as a polygon, and the objects within this polygon are defined as objects within the perception range. Using the objects within this perception range, we can construct object-centric features.
(6)$$g_{i}^{p_{i}}=\left\{r*rot\left(-\frac{w}{2}\right)*h_{i}^{p_{i}},\ldots ,h_{i}^{p_{i}},\ldots ,r*rot\left(\frac{w}{2}\right)*h_{i}^{p_{i}},l_{i}^{p_{i}}\right\}$$
(7)$$O^{\mathrm{HD}}=\left\{o_{1}^{\mathrm{HD}},o_{2}^{\mathrm{HD}},\ldots ,o_{n_{o}}^{\mathrm{HD}}\right\},\;o_{i}^{\mathrm{HD}}=\left\{o_{i,x}^{\mathrm{HD}},o_{i,y}^{\mathrm{HD}},o_{i,c}^{\mathrm{HD}}\right\}$$
(8)$$o_{i}^{p_{i}}=\left\{o_{j}^{\mathrm{HD}}|o_{j}^{\mathrm{HD}}\in g_{i}^{p_{i}}\right\}$$

Equation (Equation 6) presents polygon coordinates for the perception coverage of each link point, as shown in Figure 10. Here, *r* and *w* are the perception radius and the perception angle, respectively. The perception radius means the maximum coverage length, and the perception angle means the maximum coverage angle based on the perception coverage. Equation (Equation 7) shows the registered objects include additional objects described in Section 3.3. As shown here, each object has its x, y, and c: position x, position y, and class, respectively. Equation (Equation 8) means that the objects $o_{i}^{p_{i}}$ included in the polygon $g_{i}^{p_{i}}$. These objects can be easily extracted by their coordinates, whether the objects in the HD map are in the perception coverage polygon.

#### 4.1.4. Compose Features Based On Link Points

We have completed the preparations for constructing object-centric features through the preceding processes. To achieve more accurate estimation, we performed link interpolation, calculated the heading for each link point, and virtually formed perception coverage based on each link point, considering the sensor coverage of the target vehicle. This coverage allowed us to distinguish registered objects within this virtual perception range. The considerations when generating features in this paper include the following:Constructing unique features for each link point.Creating features that reflect object information as much as possible.Ensuring simplicity in the feature structure.

For smooth place recognition, features need to have unique characteristics, and when constructing features based on objects, the structure of the features should be as simple as possible to minimize computational complexity when comparing features.

To achieve this, we utilized a polar coordinate system to construct object information as features, as shown in the example of Figure 11. The reason for transforming the object information from the conventional Cartesian coordinate system to the polar coordinate system is that it allows the expression of object information while considering both the position of each link point and its associated heading vector. In other words, we constructed a polar coordinate system for each link point with the heading vector as the reference axis and represented the object’s information based on this system. As we already know, coordinates in polar representation are expressed as (r,θ), where θ increases counterclockwise with the heading vector set as the reference at 0 degrees. As shown in Figure 11, the object’s position can be expressed in polar coordinates for each link point as (r,θ), which can be straightforwardly formulated using the equation below.
(9)$$o_{i,(x,y)}^{p_{i}\prime }=o_{i,(x,y)}^{p_{i}}-l_{i,(x,y)}^{p_{i}}$$
(10)$$f_{i,r}^{p_{i}}=∥o_{i,(x,y)}^{p_{i}\prime }∥$$
(11)$$f_{i,\theta }^{p_{i}}=\arccos\frac{h_{i}^{p_{i}}\cdot o_{i,(x,y)}^{p_{i}\prime }}{|h_{i}^{p_{i}}\left|\right|o_{i,(x,y)}^{p_{i}\prime }|}$$
(12)$$f_{i,c}^{p_{i}}=o_{i,c}^{p_{i}}$$
(13)$$f_{i}^{p_{i}}=\left\{f_{i,r}^{p_{i}},f_{i,\theta }^{p_{i}},f_{i,c}^{p_{i}}\right\}$$

As shown in Equation (Equation 9), we can easily obtain the relative position vectors of the objects. We can also obtain its radius $f_{i,r}^{p_{i}}$ on the polar coordinate using Euclidean distance, as shown in Equation (Equation 10). Because we already obtained the relative position vectors $o_{i,(x,y)}^{p_{i}\prime }$, the theta can also easily be calculated by Equation (Equation 11), between the angle from the heading $h_{i}^{p_{i}}$. The class information is not being changed, even in the polar coordinate system, so use this information, as shown in Equation (Equation 12). Then, we finally composed our object-centric feature $f_{i}^{p_{i}}$, as shown in Equation (Equation 13).

### 4.2. Object-Centric Place Recognition

As mentioned earlier, the proposed method in this paper involves a step-by-step process of place recognition followed by pose estimation. Additionally, this method includes a continuous estimation process that leverages the previously estimated results for subsequent estimations, as shown in Figure 12; three inputs are required for place recognition. The first is the estimated pose of the vehicle from the previous frame. When a vehicle continuously moves, its location sequentially transitions from the previous place to the next. Therefore, having information about the previous place enables predictions for the next place. The second input is the object recognition results, which refer to the outcomes obtained from object recognition on sensor data. In this paper, the object recognition results assume the relative position of recognized objects to the vehicle, the type of objects, and the object recognition rate. The last input is backward information. The pose of the vehicle we want to estimate is primarily based on the forward motion of the vehicle concerning the forward direction, unless the vehicle is reversing on the road. However, suppose the estimated pose of the vehicle is mistakenly ahead of the actual location of the vehicle. In that case, we need to consider the rear location of the vehicle to correct this estimation. Backward information is the input for this purpose, and it contains two values: a value indicating whether to consider the rear link point, and a value indicating how many link points to consider as candidates in the forward or backward direction.

The place recognition module’s first task is to find the link point closest to the previous pose estimation result. Then, based on this link point, it extracts features for candidate link points that can be the next place. These features are compared with those constructed from the object recognition results to identify the link point with the most similar features. Subsequent sections will explain how candidate link points and features are obtained and compared with object recognition features.

#### 4.2.1. Extract Candidate Features

We need to limit the number of features to compare because we cannot compare all the features from link points. Here, Figure 13 presents the process to get these limited features.
(14)$$e^{t-1}=\left\{e_{x}^{t-1},e_{y}^{t-1},e_{\theta }^{t-1}\right\}$$
(15)$${\left(i,j\right)}_{t-1}=\underset{i,j}{arg min}∥l_{i}^{p_{j}}-e_{x,y}^{t-1}∥$$

As previously explained, we must first identify a set of candidates for the current place to determine the vehicle’s current pose based on the previously estimated pose. These candidates are selected based on the link point closest to the previously estimated position. Equations (Equation 14) and (Equation 15) illustrate the process of obtaining the link index $i_{t-1}$ and link point index $j_{t-1}$ closest to the previously estimated position. Here, $e^{t-1}$ is the previously estimated pose and $e_{x,y}^{t-1}$ is its position.
(16)$$C=\left\{\begin{array}{cc} \{j_{t-1},\ldots ,j_{t-1}+q\}, & b==0\\ \{j_{t-1}-b,\ldots ,j_{t-1}-b+q\}, & \mathrm{else} \end{array}\right.$$

Here, Figure 14 presented a sample of the candidate feature extraction. Generally assuming that the vehicle travels along the direction defined by the link, the order of link point indices is sequential in the direction of travel, allowing the selection of candidate link points as shown in the first condition of Equation (Equation 16). Here, *C* is the index of the candidate link points, and *b* is a backward value, where 0 selects candidate link points in the forward direction and else selects those in the opposite direction. Additionally, *q*, which will be explained later in this context, determines the range of candidate points.
(17)$$f^{C}=\left\{f^{c_{1}},f^{c_{2}},\ldots ,f^{c_{n_{c}}}\right\}$$

Moreover, the link information, including R_LinkID, L_LinkID, FromNodeID, and ToNodeID, is employed here. It is impossible to obtain *q* candidate points in any direction from the link point index $j_{t-1}$ based on the link index $i_{t-1}$. In that case, the information from FromNodeID or ToNodeID is used to fetch the deficient link points from the previous or next link. Furthermore, if link $i_{t-1}$ has R_LinkID or L_LinkID information, considering momentary lane changes during vehicle operation, the closest link point index $j_{t-1}$ based on each R_LinkID or L_LinkID is determined. As the same method to obtain candidates for the original closest link point, *q* candidate points are fetched. Combining all the obtained candidate points, the feature corresponding to each link point is retrieved, and a candidate feature $f^{C}$ is constructed, as shown in Equation (Equation 17).

#### 4.2.2. Feature Comparison

After gathering all candidate features to compare with the object recognition results, the remaining task is identifying which candidate feature is most similar to the features derived from object recognition results, as shown in Figure 15. To perform the comparison, we must compose the recognition feature from the object recognition results. By comparing this recognition feature with all the candidate features, a specific feature with the slightest difference or the most similarity is selected, and its link point following the particular feature is the desired place for the recognition result.
(18)$$f^{R}=\left\{f_{1}^{R},f_{2}^{R},\ldots ,f_{n_{r}}^{R}\right\}$$
(19)$$f_{i}^{R}=\left\{f_{i,r}^{R},f_{i,\theta }^{R},f_{i,c}^{R},f_{i,p}^{R}\right\}$$

As mentioned in Section 3.2, the recognition features are generated as shown in Figure 16. The recognition feature $f^{R}$ is constructed using object recognition results containing one more piece of information for each object $f_{i}^{R}$, as shown in Equations (Equation 18) and (Equation 19). Unlike features constructed using existing HD map information, one more information is the confidence from object recognition $f_{i,p}^{R}$.
(20)$$∥f_{k}^{R}-f_{j}^{c_{i}}∥=\left\{\begin{array}{l} \begin{array}{c} \sqrt{\begin{array}{cc} \; & {\left(f_{k,r}^{R}\right)}^{2}+{\left(f_{j,r}^{c_{i}}\right)}^{2}\\ \; & -2*f_{k,r}^{R}*f_{j,r}^{c_{i}}\\ \; & *\cos f_{k,\theta }^{R}-f_{j,\theta }^{c_{i}} \end{array}} \end{array},\begin{array}{cc} \; & \mathrm{if}\;f_{k,c}^{R}=f_{j,c}^{c_{i}}\;\mathrm{and}\\ \; & f_{k,p}^{R}>\tau_{p} \end{array}\\ r,\;\mathrm{else} \end{array}\right.$$

Equation (Equation 20) illustrates how the elements of the recognition feature $f^{R}$ and the candidate feature $f^{C}$ are compared. In this equation, $f_{k}^{R}$ is the *k*-th element of $f_{R}$, and $f_{j}^{c_{i}}$ is the *j*-th element of *i*-th candidate feature. This paper employs the Euclidean distance within the polar coordinates to measure the similarity between two objects. A smaller value indicates a higher similarity between the two elements. This comparison incorporates the objects’ (r,θ) values and factors in the object class and recognition confidence. As depicted in Equation (Equation 20), the Euclidean distance is only computed when the classes of the two elements are the same or when the confidence exceeds a specific threshold, denoted as $\tau_{p}$. Otherwise, the comparison value between the two elements is fixed at the vehicle’s maximum detection range, denoted as *r*.
(21)$$i^{*}=\underset{i}{arg min}\sum_{k}^{n_{r}}\min_{\forall j\in n_{c_{i}}}∥f_{k}^{R}-f_{j}^{c_{i}}∥$$
(22)$$l_{t_{i}^{*}}^{p_{t_{j}^{*}}}=f^{c_{i^{*}}}$$

Using Equation (Equation 20), a formula for determining the index of the most similar feature among all candidate features can be constructed, as shown in Equation (Equation 21). Here, $n_{r}$ is the number of feature elements of $f_{R}$, $n_{c_{i}}$ is the number of feature elements of the candidate *C*, and $i^{*}$, $t_{i}^{*}$, $t_{j}^{*}$ is the optimal matched candidate index, link index at time *t*, link point index at time *t*, respectively. As evident from the equation, it compares the elements of the recognition feature with those of each candidate feature, and the candidate feature with the smallest sum represents the feature corresponding to the desired place. Equation (Equation 22) indicates the link point corresponding to the feature index obtained from this formula.

### 4.3. Object-Centric Pose Estimation

Through the above place recognition process, we can confirm that the vehicle is traversing a specific location. However, what we desire is not place recognition, but rather the estimation of the vehicle’s pose. Recognizing that the vehicle is near a specific place, i.e., a particular link point, can be determined through the abovementioned process. However, this does not imply precision in the vehicle’s pose. The confirmation provided by place recognition indicates that the vehicle is driving closest to this place. However, utilizing the results of this place recognition enables the estimation of the vehicle’s precise pose.

As illustrated in Figure 17, object-centric pose estimation involves three types of inputs. The first input is the information about the recognized place from the preceding stage, and the second is the recognition feature already constructed in the previous stage. The final input is the vehicle’s control input, which is crucial in pose estimation.

This module incorporates an extended Kalman filter (EKF) for pose estimation. The EKF takes the vehicle’s control input as input and performs prediction. Since the control input provided by the vehicle is considerably faster in frequency than object recognition, the EKF generally performs predictions based on control input alone. However, when information for pose estimation is obtained through object and place recognition, it incorporates this information for an update. Because the recognition process is not always performed, additional information from the RTK sensor, camera odometry, and lidar odometry is used for updates when this information is available, as represented in Figure 17. In the following sections, we will demonstrate how object and place recognition information are used to achieve more precise vehicle pose estimation. Additionally, we will explain the modified EKF used in this paper and elucidate how the backward information used in place recognition is generated.

#### 4.3.1. Object-Centric Pose Estimation

We know the vehicle is near a specific location, since we have already performed place recognition in the preceding stage. While each link point within that place has its position and direction, in reality, the vehicle does not precisely align with the location and direction of most link points. Although each link point represents the path the vehicle would ideally take along the lane, it is almost impossible for a human or autonomous vehicle to travel exactly along this ideal path. Therefore, to accurately estimate the vehicle’s position, we must determine how far away and rotated it is from the link point location obtained through place recognition.

Suppose the results of object recognition obtained from the vehicle are perfect, without any errors, and the vehicle also travels in an ideal manner, perfectly aligning with the positions and directions of the link points. In this scenario, if we use the recognition feature extracted from the vehicle’s object recognition results to represent objects based on the position and direction of the link point, it would precisely overlap with the information of objects registered in the semantic HD map. Leveraging this characteristic, we aim to calculate how far and rotated the current vehicle is from the link point.

Considering that object recognition by the vehicle has some errors, but is reliable enough to perform place recognition with a sufficient number of objects and recognition rate, we can obtain the optimal link point through place recognition. Then, following the process based on the assumption made earlier, we use the recognition feature to represent objects based on the position and direction of the link point obtained from the place recognition results. Naturally, those represented recognized objects will be located at positions different from those registered in the existing semantic HD map. However, unless the errors in the object recognition results are substantial, the represented objects should be at a similar distance and direction compared to the existing semantic HD map objects, as shown in Figure 18. Suppose we adjust the represented objects obtained through the recognition feature to match the existing semantic HD map objects as closely as possible. In that case, the movement of distance and direction is generated, and we can estimate the vehicle’s pose using this movement to adjust with the link point from the place recognition result. This re-projection feature process is shown in Figure 17.

Assume that the re-projected objects using the recognition feature based on the best link point as:(23)$$O_{R}=\left\{o_{r_{1}},o_{r_{2}},\ldots ,o_{r_{n_{r}}}\right\}$$
(24)$$o_{r_{i}}=\left\{o_{r_{i},x},o_{r_{i},y},o_{r_{i},c},o_{r_{i},p}\right\}$$

As shown in Equations (Equation 23) and (Equation 24), the recognition features are re-projected in the Cartesian coordinate system, based on the link point $l_{t_{i}^{*}}^{p_{t_{j}^{*}}}$.

Because we need to find the best transform $T^{*}$ between HD objects and re-projected objects, here, we used a class-based iterative closest points (ICP), which modified ICP for this paper, as shown in Figure 19. The objects included in the link point $l_{t_{i}^{*}}^{p_{t_{j}^{*}}}$ can be expressed as:(25)$$\mathrm{simply},\;i^{*}=t_{i}^{*},\;j^{*}=t_{j}^{*},$$
(26)$$o_{i^{*}}^{p_{j^{*}}}=\left\{o_{i^{*},1}^{p_{j^{*}}},o_{i^{*},2}^{p_{j^{*}}},\ldots ,o_{i^{*},n_{l_{i^{*}}}}^{p_{j^{*}}}\right\}$$
(27)$$o_{i^{*},k}^{p_{j^{*}}}=\left\{o_{i^{*},\left(k,x\right)}^{p_{j^{*}}},o_{i^{*},\left(k,y\right)}^{p_{j^{*}}},o_{i^{*},\left(k,c\right)}^{p_{j^{*}}}\right\}$$

Then, we need to find matched pairs between $O^{R}$ and $o_{i^{*}}^{p_{j^{*}}}$, considering their position, class, and abjectness. Similar with Equations (Equation 20) and (Equation 21), we composed class-based matched pairs as:(28)$$M=\{\left(o_{r_{1}},o_{m_{1}}\right),\left(o_{r_{2}},o_{m_{2}}\right),\ldots ,\left(o_{r_{n_{r}}},o_{m_{n_{r}}}\right)\}$$
(29)$$o_{m_{q}}=\underset{o_{i^{*},k}^{p_{j^{*}}}}{arg min}∥o_{m_{q},\left(x,y\right)}-o_{i^{*},k,\left(x,y\right)}^{p_{j^{*}}}∥,\;\begin{array}{cc} \; & o_{m_{q},c}==o_{i^{*},k,c}^{p_{j^{*}}}\;\\ \; & \mathrm{and}\;\;o_{m_{q},c}>\tau_{p} \end{array}$$

Now, we can solve the ICP problem for the matched pair *M* as shown below:(30)$$T^{*}=\underset{T}{arg min}\frac{1}{n_{r}}\sum_{i}^{n_{r}}∥T\;o_{r_{i},\left(x,y\right)}-o_{m_{i},\left(x,y\right)}∥$$
(31)$$T=\left[\begin{array}{ccc} \cos\theta_{t} & -\sin\theta_{t} & x_{t}\\ \sin\theta_{t} & \cos\theta_{t} & y_{t} \end{array}\right]$$

From Equation (Equation 31), we can find optimal transformation $T^{*}$ and its parameter $\left\{x_{t}^{*},y_{t}^{*},\theta_{t}^{*}\right\}$. Using this optimal solution, we can finally estimate vehicle pose at *t* frame as:(32)$$s^{t}=\left[\begin{array}{c} l_{i^{*},x}^{p_{j^{*}}}-x_{t}^{*}\\ l_{i^{*},y}^{p_{j^{*}}}-y_{t}^{*}\\ l_{i^{*},\theta }^{p_{j^{*}}}-\theta_{t}^{*} \end{array}\right]$$

Note that this estimated pose $s^{t}$ is used to update EKF, as shown in Figure 17. Also, this information is used to check backwardness, as described below in more detail.

#### 4.3.2. Modified Extended Kalman Filter (EKF)

While it could be sufficient to use the results obtained in Section 4.3.1 directly for object-centric pose estimation, estimating the actual pose of the vehicle has to be performed continuously at a high frequency, ideally. However, the sensors primarily used for object recognition, which is the basis of the object-centric approach, include cameras, LiDAR, and radar, among others. Among these sensors, cameras typically acquire data at around 30 fps, LiDAR at 15 fps, and radar at approximately 50 fps. Assuming each sensor’s data are synchronized, the data cycle is roughly 15 to 30 fps based on the object recognition result. However, the fps may vary depending on the choice of object recognition algorithm.

For a typical vehicle, even if it moves at a low speed, it is generally faster than a person walking. Assuming highway driving at 100 km/h, the vehicle would move more than 27 m per second. Therefore, estimating the actual pose of the vehicle needs to be performed much faster than the fps at which sensor data are acquired for object recognition. In this paper, to address this, we use an extended Kalman filter (EKF) to predict the vehicle’s pose during periods. For prediction, the control input includes the vehicle’s speed and steering information, directly obtainable from the vehicle’s CAN communication. Since the vehicle’s CAN communication has a frequency of 200 Hz or higher, pose prediction can be conducted at a faster rate.

If object recognition can find all referenced objects inside each frame for all frames, we can trust these results. As mentioned above, object recognition is not guaranteed a specific frame rate. Furthermore, object recognition is not always performed successfully to find objects for all input camera frames. To solve this problem, we composed our EKF with additional information: RTK, camera odometry, and lidar odometry, as shown in Figure 17. This information supports the pose estimation process when the object recognition is uncertain and ambiguous. Note that this modified EKF is designed to be capable of various methods for additional information. Any RTK sensor, camera, and lidar odometry method can adjusted for this.
(33)$${\vec{x}}_{k}={\left(x_{k}\;y_{k}\;\theta_{k}\;{\dot{x}}_{k}\;{\dot{y}}_{k}\;{\dot{\theta }}_{k}\right)}^{t}$$
(34)$$u_{k}={\left(v_{k}\;\delta_{k}\right)}^{t}$$
(35)$${\hat{\vec{x}}}_{k}^{¯}=f\left({\hat{\vec{x}}}_{k-1},u_{k}\right)$$
(36)$${P_{k}}^{\;¯}=A_{k}P_{k-1}A_{k}^{T}+Q_{k}$$
(37)$$K_{k}={P_{k}}^{\;¯}H_{k}^{T}{\left(H_{k}{P_{k}}^{\;¯}H_{k}^{T}+R_{k}\right)}^{-1}$$
(38)$${\hat{\vec{x}}}_{k}={\hat{\vec{x}}}_{k}^{¯}+K_{k}\left(z_{k}-h\left({\hat{\vec{x}}}_{k}^{¯}\right)\right)$$
(39)$$P_{k}={P_{k}}^{\;¯}-K_{k}H_{k}{P_{k}}^{\;¯}$$

As mentioned above, we estimated the vehicle’s pose through the EKF. In this paper, we define the state vector ${\vec{x}}_{k}$ and control vector $u_{k}$ using Equations (Equation 33) and (Equation 34), respectively. Here, ${\vec{x}}_{k}$ represents the state of the vehicle to be estimated at time *k*, and $u_{k}$ represents the speed and rotation of the vehicle at time *k*. As shown in Equations (Equation 35) and (Equation 36), we predict ${\vec{x}}_{k}$ through vehicle kinematics in Figure 20, estimate ${\vec{x}}_{k}$ using the Kalman gain *K* and measurement vector $z_{k}$, and update the process covariance matrix *P*.
(40)$$\begin{array}{cc} {\hat{\vec{x}}}_{k}^{¯} & =f({\hat{\vec{x}}}_{k-1},u_{k})\\ \left[\begin{array}{c} {\hat{x}}_{k}^{¯}\\ {\hat{y}}_{k}^{¯}\\ {\hat{\theta }}_{k}^{¯}\\ {\hat{\dot{x}}}_{k}^{¯}\\ {\hat{\dot{y}}}_{k}^{¯}\\ {\hat{\dot{\theta }}}_{k}^{¯} \end{array}\right] & =\left[\begin{array}{c} {\hat{x}}_{k-1}\\ {\hat{y}}_{k-1}\\ {\hat{\theta }}_{k-1}\\ {\hat{\dot{x}}}_{k-1}\\ {\hat{\dot{y}}}_{k-1}\\ {\hat{\dot{\theta }}}_{k-1} \end{array}\right]+\left[\begin{array}{c} v_{k}\cos\left({\hat{\theta }}_{k-1}+\delta_{k}\right)\cdot \Delta t\\ v_{k}\sin\left({\hat{\theta }}_{k-1}+\delta_{k}\right)\cdot \Delta t\\ v_{k}\sin(\delta_{k})\;/\;L\cdot \Delta t\\ v_{k}\cos\left({\hat{\theta }}_{k-1}+\delta_{k}\right)\\ v_{k}\sin\left({\hat{\theta }}_{k-1}+\delta_{k}\right)\\ v_{k}\sin(\delta_{k})\;/\;L \end{array}\right] \end{array}$$
(41)$$\begin{array}{cc} A_{k} & =\frac{\partial f}{\partial x}{|}_{x={\hat{\vec{x}}}_{k-1}}\\ \; & =\left[\begin{array}{cccccc} 1 & 0 & -v_{k}\sin\left({\hat{\theta }}_{k-1}+\delta_{k}\right)\cdot \Delta t & 0 & 0 & 0\\ 0 & 1 & v_{k}\cos\left({\hat{\theta }}_{k-1}+\delta_{k}\right)\cdot \Delta t & 0 & 0 & 0\\ 0 & 0 & 1 & 0 & 0 & 0\\ 0 & 0 & -v_{k}\sin\left({\hat{\theta }}_{k-1}+\delta_{k}\right) & 1 & 0 & 0\\ 0 & 0 & v_{k}\cos\left({\hat{\theta }}_{k-1}+\delta_{k}\right) & 0 & 1 & 0\\ 0 & 0 & 0 & 0 & 0 & 1 \end{array}\right] \end{array}$$

In Equation (Equation 40), we used vehicle kinematics as shown in Figure 20 to obtain the predicted state vector ${\hat{\vec{x}}}_{k}^{¯}$. Because we used the EKF for non-linear estimation in this paper, we calculated the state transition matrix $A_{k}$ by expressing it as a Jacobian matrix of ${\hat{\vec{x}}}_{k-1}$ for the function *f*, as in Equation (Equation 41).
(42)$$\begin{array}{cc} z_{k} & =h({\hat{\vec{x}}}_{k}^{¯})\\ \left[\begin{array}{c} x_{rtk}\\ y_{rtk}\\ \Delta x_{camera}\\ \Delta y_{camera}\\ \Delta \theta_{camera}\\ \Delta x_{lidar}\\ \Delta y_{lidar}\\ \Delta \theta_{lidar} \end{array}\right] & =\left[\begin{array}{c} {\hat{x}}_{k}^{¯}-L_{f}\cos{\hat{\theta }}_{k}^{¯}\\ {\hat{y}}_{k}^{¯}-L_{f}\sin{\hat{\theta }}_{k}^{¯}\\ {\hat{x}}_{k}^{¯}-{\hat{x}}_{k-1}\\ {\hat{y}}_{k}^{¯}-{\hat{y}}_{k-1}\\ {\hat{\theta }}_{k}^{¯}-{\hat{\theta }}_{k-1}\\ {\hat{x}}_{k}^{¯}-{\hat{x}}_{k-1}\\ {\hat{y}}_{k}^{¯}-{\hat{y}}_{k-1}\\ {\hat{\theta }}_{k}^{¯}-{\hat{\theta }}_{k-1} \end{array}\right] \end{array}$$
(43)$$\begin{array}{cc} H_{k} & =\frac{\partial h}{\partial x}{|}_{x={\hat{\vec{x}}}_{k}^{¯}}\\ \; & =\left[\begin{array}{cccccc} 1 & 0 & L_{f}\sin{\hat{\theta }}_{k}^{¯} & 0 & 0 & 0\\ 0 & 1 & -L_{f}\cos{\hat{\theta }}_{k}^{¯} & 0 & 0 & 0\\ 1 & 0 & 0 & 0 & 0 & 0\\ 0 & 1 & 0 & 0 & 0 & 0\\ 0 & 0 & 1 & 0 & 0 & 0\\ 1 & 0 & 0 & 0 & 0 & 0\\ 0 & 1 & 0 & 0 & 0 & 0\\ 0 & 0 & 1 & 0 & 0 & 0 \end{array}\right] \end{array}$$

The measurement vector $z_{k}$ used in this paper is given by Equation (Equation 42). The $z_{k}$ is defined for the additional information, as shown in Figure 21. As shown in Figure 21, the RTK sensor data are directly used for the measurement with only concern about its placement. The camera odometry and lidar odometry measurements are defined here as differences in states from the previous time to the current time. Note that the states for these odometry measurements are *x*, *y*, and θ. Similar to $A_{k}$, the measurement transition matrix $H_{k}$ of the EKF is determined by expressing the Jacobian matrix of ${\hat{x}}_{k}^{¯}$ for the function *h*, as shown in Equation (Equation 43).

In this way, we have constructed our modified EKF. However, in actual application, the cycle of all sensor inputs and estimation results are different, and the cycle may also change momentarily depending on the system’s stability. If we wait for all data to be input, not only will the measurement update be slow, but the time of each data will be different so that errors may occur during the update. Therefore, the measurement vector configured above is used as is, but the update part is configured to operate each time each sensor and estimation result data are acquired.

#### 4.3.3. Check Backwardness

The backward information, which serves as the input for Section 4.2.1 in the object-centric place recognition, is determined right here. As mentioned earlier, this process is information includes the variables *b*, which decides whether to consider the front or rear links based on the current position, and *q* determines how many candidate links to retrieve. Since vehicles typically move forward according to the predefined direction of the road, it is usually sufficient to focus on the front links for estimation. However, the estimated position may be ahead of the actual vehicle location in cases where the estimation is incorrect. If this error accumulates, it can significantly reduce the accuracy of the pose estimation module. To prevent these unexpected cases, we defined a specific range of rear links included in the candidates for consideration in the estimation process.
(44)$$b=\left\{\begin{array}{cc} \mathrm{int}(\left(v_{t}*\delta_{r}-∥s_{x,y}^{t}-\left({\hat{x}}_{t}^{¯},{\hat{y}}_{t}^{¯}\right)∥\right)\;/\;\tau_{i}), & |\arctan\left(s_{x,y}^{t}-\left({\hat{x}}_{t}^{¯},{\hat{y}}_{t}^{¯}\right)\right)-{\hat{\theta }}_{t}^{¯}|>\frac{2}{\pi }\\ 0, & else \end{array}\right.$$

In Equation (Equation 44), $\delta_{r}$ represents the periodicity of object recognition results, and $\tau_{i}$ denotes the interpolation interval in Section 4.1.1. As seen here, the value of *b* is determined by the difference between the estimation in Section 4.3.1 and the prediction in Section 4.3.2. Here, the value is determined looking back by the amount of the positional difference between the two results when the difference in their angles exceeds $\frac{2}{\pi }$.
(45)$$q=\max(\mathrm{int}\left(\left(v_{t}*\delta_{r}-∥s_{x,y}^{t}-\left({\hat{x}}_{t}^{¯},{\hat{y}}_{t}^{¯}\right)∥\right)\;/\;\tau_{i}\right)+\tau_{m},\tau_{m})$$

The $\tau_{m}$ in Equation (Equation 45) is determined as a hyperparameter representing the minimum number of links to be considered for place recognition. The equation shows that the number of candidate links from the minimum link count increases by the positional difference between the results from Section 4.3.1 and Section 4.3.2. By incorporating this straightforward value, the uncertainty of the estimation can be effectively compensated in this paper.

## 5. Experiments

### 5.1. Simulation

Conducting experiments is essential to validate the object-centric pose estimation method outlined earlier. In practical application on actual vehicles, preparations such as sensor procurement and installation, vehicle modification, and environmental setup require significant time and cost. Therefore, before conducting experiments in a real-world scenario, initially confirming the method’s effectiveness through simulation is more efficient.

Constructing a simulation environment involves creating an idealized experimental setting resembling or comparable to the natural environment. The ideal simulation method involves

Placing structures such as roads and buildings;Generating virtual vehicles;Creating synthetic sensor data;Applying object recognition algorithms to the virtual data;Confirming the validity of this paper’s method through the obtained results.

This simulation method is ideal and nearly directly applicable to a real-world environment.

However, this approach demands considerable time and effort, akin to the effort required for setting up experiments in a natural environment, if not more. As the simulation environment approaches the realism of the actual scenario, it requires more meticulous work. Therefore, the simulation in this paper has been intentionally kept as simple as possible, focusing solely on confirming the validity of the proposed method.

To verify the method presented in this paper, acquiring and processing actual sensor data or obtaining results through object recognition is unnecessary. As mentioned earlier, the method assumes that such results are already available for object-centric pose estimation. Hence, as long as synthetic object recognition results can be generated from simulation, experimenting with this paper’s method poses no challenges. Additionally, even without constructing an actual vehicle, simulation execution is possible by providing virtual control input that the vehicle typically delivers. Therefore, this paper establishes a straightforward simulation environment and demonstrates the method’s efficacy through experimentation.

#### 5.1.1. Target Environment

As mentioned, we already have the actual environment targeted for our experiments. Consequently, we possess an HD map for this environment. This real-world environment includes challenging conditions such as bridges, the undersides of overpasses, tunnels, and other locations where RTK sensor signals are vulnerable. Due to these conditions, our environment is suitable for experimenting with the efficacy of the proposed method in this paper. As discussed in Section 3, we constructed a semantic HD map by additionally registering objects mentioned in Section 3.3 to facilitate the desired object-centric pose estimation. The total number of initially constructed and additionally added objects amounted to approximately 5000, and detailed numbers for each object are shown in Table 7.

Furthermore, the procedures outlined in Section 4.1 were executed to configure the environment. Initially, link interpolation in Section 4.1.1 was performed with a $\tau_{i}$ value of 1 m. Subsequently, as described in Section 4.1.2 and Section 4.1.4, object-centric features for all link points were precomputed and stored in the semantic HD map. Given that the dimensionality of the features is not substantial, we opted to store the feature values themselves without the need for additional compression processes, as it did not impose significant storage requirements.

As emphasized, we aimed to keep the simulation environment concise. After completing all the steps mentioned above on the semantic HD map derived from the existing HD map, we configured the environment for our desired simulation, as illustrated in Figure 22.

#### 5.1.2. Imaginary Object Recognition

Now that the target environment has been fully configured, the crucial task is determining how to generate Imaginary Object Recognition results. In the preceding preparation steps, we have already implemented object-centric features for each link, enabling us to construct Imaginary Object Recognition results similarly.
(46)$$\begin{array}{cc} l_{i,x}^{p_{i}\prime } & =l_{i,x}^{p_{i}}+\epsilon_{l_{x}},\;\epsilon_{l_{x}}∼N\left(0,\sigma_{l_{x}}^{2}\right)\\ l_{i,y}^{p_{i}\prime } & =l_{i,y}^{p_{i}}+\epsilon_{l_{y}},\;\epsilon_{l_{y}}∼N\left(0,\sigma_{l_{y}}^{2}\right)\\ h_{i}^{p_{i}\prime } & =h_{i}^{p_{i}}+\epsilon_{l_{h}},\;\epsilon_{l_{h}}∼N\left(0,\sigma_{l_{h}}^{2}\right) \end{array}$$
(47)$$\begin{array}{cc} o^{R} & =\{o_{1}^{R},o_{2}^{R},\ldots ,o_{nl_{i}}^{R}\}\\ o_{k}^{R} & =\{o_{k,x}^{R},o_{k,y}^{R},o_{k,c}^{R},o_{k,p}^{R}\}\\ o_{k,x}^{R} & =o_{i,(k,x)}^{p_{i}}+\epsilon_{o_{x}},\;\epsilon_{o_{x}}∼N\left(0,\sigma_{o_{x}}^{2}\right)\\ o_{k,y}^{R} & =o_{i,(k,y)}^{p_{i}}+\epsilon_{o_{y}},\;\epsilon_{o_{y}}∼N\left(0,\sigma_{o_{y}}^{2}\right)\\ o_{k,c}^{R} & =o_{i,(k,c)}^{p_{i}}\\ o_{k,p}^{R} & =min\left(1,1+\epsilon_{o_{p}}\right),\;\epsilon_{o_{p}}∼N\left(0,\sigma_{o_{p}}^{2}\right) \end{array}$$

As evident in Equations (Equation 46) and (Equation 47), we begin by assuming the existence of a specific link point and generating information for a new point by introducing a random offset relative to this reference point. Additionally, the positions of objects corresponding to the link point are created with similar offsets. Importantly, we need to include one more piece of information: the recognition rate of the objects. The recognition rate is determined as shown in Equation (Equation 47).

This approach can generate Imaginary Object Recognition features for each link point, as depicted in Figure 23. Based on the link point that shifts with the vehicle’s movement while driving, we generate virtual imaginary object recognition and its associated features each time the car progresses.

#### 5.1.3. Simulation

Having configured the target environment and generated Imaginary Object Recognition and its associated features based on it, what remains is to simulate the vehicle’s virtual driving within the simulation to confirm the effectiveness of the proposed method in estimating the vehicle’s pose. In this simulation experiment, we fixed the driving path to cover the entire section shown in Figure 22 continuously while varying the conditions during the experiment. This path includes a bridge and a tunnel, particularly a long bridge, enabling experimentation in areas where RTK sensor signals are not easily accessible.

Even when driving the same path within the simulation, we conducted experiments by changing conditions to test whether the proposed method in this paper exhibits high accuracy under various condition variations. Although not explicitly mentioned in Section 4.3, as the proposed method is fundamentally object-centric when the RTK sensor signals are robust, this information is used as an update input for the EKF in estimating the vehicle’s pose.

The hyperparameters varied during this simulation mainly include $\lambda_{l_{x}}$, $\lambda_{l_{y}}$, $\lambda_{l_{h}}$, $\lambda_{o_{x}}$, $\lambda_{o_{y}}$, $\lambda_{o_{p_{\min}}}$, and $\lambda_{o_{p_{\max}}}$. The parameters r=100 m, $w=60^{\circ }$, $\tau_{p}=0.5$, $\tau_{i}=1$ m, and $\tau_{m}=10$ were fixed throughout the simulation experiments. As testing all possible combinations of the hyperparameters mentioned above is practically infeasible, we conducted experiments for the following three cases.

Ideal case:$\sigma_{l_{x}}=0.0,\;\sigma_{l_{y}}=0.0,\;\sigma_{l_{h}}=0.0,\;\sigma_{o_{x}}=0.0,\;\sigma_{o_{y}}=0.0,\;\sigma_{o_{p}}=0.0$.In the case of slight errors:$\sigma_{l_{x}}=0.15,\;\sigma_{l_{y}}=0.15,\;\sigma_{l_{h}}=3.0,\;\sigma_{o_{x}}=0.15,\;\sigma_{o_{y}}=0.15,\;\sigma_{o_{p}}=0.3$.In the case of relatively pronounced errors:$\sigma_{l_{x}}=0.3,\;\sigma_{l_{y}}=0.3,\;\sigma_{l_{h}}=6.0,\;\sigma_{o_{x}}=0.3,\;\sigma_{o_{y}}=0.3,\;\sigma_{o_{p}}=0.45$.

For the control input during the experiment, at each prediction cycle, $\delta_{e}$ of the EKF, the velocity and direction required to move from the current link point to the following link point are calculated. These values are then formulated into the control input and provided as the control input for Section 4.3.

Table 8 presents the position estimation results while receiving data from the RTK sensor. During this experiment, no information from the RTK sensor was received in the tunnel and on the bridge. Therefore, RTK-based position correction was not performed in these areas. When RTK position correction was applied, a random error within 0.05 m was added, following the RTK sensor error. As shown in Table 8, the position estimation performs remarkably well under ideal conditions. When the perception results are slightly or significantly less accurate, the correction from RTK ensures that the estimates stay consistent.

In cases where the RTK sensor data are available, Table 8 exhibited stable and accurate corrections. To compare with Table 8, here, experiments were conducted without receiving input from the RTK sensor, as shown in Table 9, relying solely on the proposed method for position estimation. As evident in Table 9, under ideal conditions, our proposed method demonstrated error levels comparable to those achieved with RTK sensor correction. Even with slight perception errors, the position estimation was relatively stable. However, the correction position was incorrect in situations where significant perception errors occurred, leading to a sharp increase in errors.

Figure 24, Figure 25 and Figure 26 illustrate the comparison of all estimated paths in ideal, slight, and pronounced erroneous perception cases without RTK information. Upon reviewing the results in Table 8 and Table 9, it is evident that the proposed method exhibits reliable performance, especially when the object recognition accuracy is high. Nevertheless, Table 9 and Figure 27 also present that the estimation performance can be ruined when the object recognition is performed poorly.

### 5.2. Actual Experiments

For the actual experiments in this paper, we prepared our vehicle with various sensors, as shown in Figure 28. We placed one RTK, four cameras, three lidars, and five radars for this vehicle. We installed radar because we can estimate the distance of objects using radar data, and this information can also be adjusted to object recognition results from cameras. As shown in Figure 29, sensor placements are composed to recognize objects around the vehicle. Moreover, we focused our sensor placements on the front side of the vehicle because the front side is the main field of view during driving. Lidar sensors are concentrated on the front side only because we intended to make dense point clouds to extract more precise features.

For the object recognition for camera input, we adjusted YOLO v8 [62] in this paper. YOLO v8 has a swift processing time, and can detect objects surprisingly well, even with few datasets. To train YOLO v8, we collected images using this vehicle and labeled objects inside each image, as shown in Table 7. We composed the dataset, which includes approximately 5000 images and object labels. Here, we used 3700 images for training and 1300 for validation.

Here, Figure 30 and Figure 31 present results for the validation set after training YOLO v8. As shown here, results are qualified with 0.962 mean average precision for all classes when Intersection over Union (IoU) is set to 0.5. However, this result does not guarantee that the object detection is well performed for the actual environments. Object detection accuracy using YOLO v8 is undoubtedly degraded in the natural environment because of its complex conditions.

As Figure 17 explained, we used RTK data, camera odometry, and lidar odometry. As said in Equation (Equation 42), RTK data can be directly adjusted. In this paper, we adjust the ORB method [63] for camera odometry and the Fast-LIO method [64] for lidar odometry. These methods ensure fast enough to achieve real-time processing and exact odometry estimation. These odometry are adjusted for the measurement update for the modified EKF, as explained in Equation (Equation 42).

For the actual experiments, the process mentioned in Section 5.1.1 is performed precisely the same way. In the actual experiments, we focused on two main areas: bridge and tunnel. Because areas other than these two can acquire precise RTK sensor data, pose estimation is also being updated precisely. However, the pose estimation will drift inside these two areas because RTK sensor data are unstable or disabled. To compare our proposed method in this paper, we generated five paths in the experiments: RTK, camera, lidar, camera+lidar, and proposed. The RTK sensor data draw the RTK path, the camera odometry draws the camera path, the lidar odometry draws the lidar path, the combination odometry of both draws the camera+lidar path, and the proposed path is drawn by this paper method.

As shown in Figure 32, we perform experiments on our proposed method in the bridge and tunnel in this paper. As explained in Section 5.1.1, the bridge and tunnel cannot acquire proper RTK sensor data because these environments are blocked with structures, as shown in Figure 32. Here, we perform manual driving while adhering to the prescribed speed limit because our proposed method can perform manual and autonomous driving. The prescribed speed limit on the bridge is set to 80 km/h and 60 km/h on the bridge and tunnel, respectively.

Figure 33 and Figure 34 are the actual experiment result paths on the bridge and tunnel. As mentioned earlier, we generated five paths for each experiment. Here, ’Merged’ presents the camera+lidar odometry path. The right of each figure presents the most distinctive section from each experiment. As shown in Figure 33 and Figure 34, all paths are estimated finely, but errors are accumulated if the driving path is more extended. As expected, the RTK path is not matched on the ground-truth path, but it does not draw the wrong path because our RTK sensor has filters. The camera and lidar odometry draw more accurate paths than expected, and the camera+lidar odometry path is more precise than any other single information. The path we proposed in this paper in these figures is even more accurate than other estimated paths and better laid on the ground truth.

Table 10 presents the longitudinal and lateral errors for all paths, compared with the ground truth path. We compare link points from the HD map with the ground truth path because we cannot acquire exact coordinates from the vehicle’s RTK sensor while driving on the bridge and tunnel. As shown in Table 10, the lidar path is entirely inaccurate compared to other paths because the frame rate of the lidar is not enough to estimate odometry in real-time for the fast velocity. The camera path is more accurate than the lidar, but still inaccurate. As we know, the features extracted from an image are changed according to the environment, and it causes unexpected estimations. Surprisingly, the RTK path is more accurate than we expected. As mentioned above, our RTK sensor has filters and estimates its pose even if the RTK signal is lost. The camera+lidar path is generated more precisely than the camera or lidar-only path. We can imagine that the camera and lidar odometry compensated with each other to derive a more accurate pose estimation. Finally, our proposed method presents even more precise results than the camera+lidar path. Although object recognition trained on very few datasets is performed on each image frame, our proposed method presents more accurate pose estimations, even in the bridge and tunnel. Our proposed method can adapt to any existing pose estimation method and perform more robust pose estimation using the object information. Also, the processing time of the proposed method in this paper is less than 100 ms for each pose estimation in the NVIDIA Orin kit, and it will reduced after optimization.

## 6. Conclusions and Future Works

This paper presents an object-centric hierarchical pose estimation method using semantic HD maps for general autonomous driving. Instead of constructing features based on conventional sensor data, we propose a method for creating conceptually more straightforward yet robust object-based features. Furthermore, based on the generated object-centric features, we introduce a hierarchical estimation method that utilizes HD map information to perform place recognition followed by pose estimation. Experimental results confirm that if the proposed method performs well in object recognition, it can achieve high accuracy. However, the proposed method has several limitations:The accuracy of object recognition heavily influences the performance of the proposed method. While this is expected, given that the technique itself is object-centric, the lack of specific measures to compensate for low object recognition accuracy can be considered a drawback.Achieving high performance is challenging when the number of objects registered in the HD map is limited. The proposed method assumes a substantial number of objects recorded in the HD map for prosperous place and pose recognition, which may require additional effort, time, and cost.If the current position is completely lost, there is no method to rediscover the position without RTK sensor inputs. This problem is usually called global re-localization, and the proposed method relies on RTK sensor inputs for correction.

Despite these challenges, the method presented in this paper holds significance, especially considering the recent advancements in deep learning networks with high object recognition accuracy. The methodology offers a relatively simple yet high-performance pose estimation approach if utilized to construct a semantic HD map.

In future research, we aim to address the limitations of the method proposed in this paper. Because HD map objects power our proposed method, adding more objects inside the existing HD map is the simplest way to adjust our method efficiently. While this paper manually conducted this task, recent research has explored methods for accurately distinguishing objects. Therefore, we will automatically register our desired object in the existing HD map from these advancements.

Furthermore, to tackle the global re-localization issue mentioned earlier, we intend to develop a method that involves reading the initial position to find an area where the vehicle might be present. If there is a specific graph in which each node consists of an object, we can divide sections using graph clustering. Then, a particular place will be in a section as a subgraph. Therefore, we can conduct sub-graph matching approaches to find a place where the vehicle is located.

## Figures and Tables

**Figure 1 sensors-24-05191-f001:**
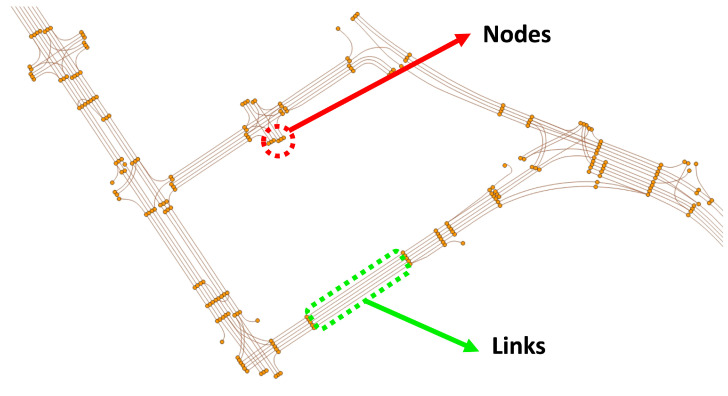
Nodes and links. Circles are nodes, and lines are links.

**Figure 2 sensors-24-05191-f002:**
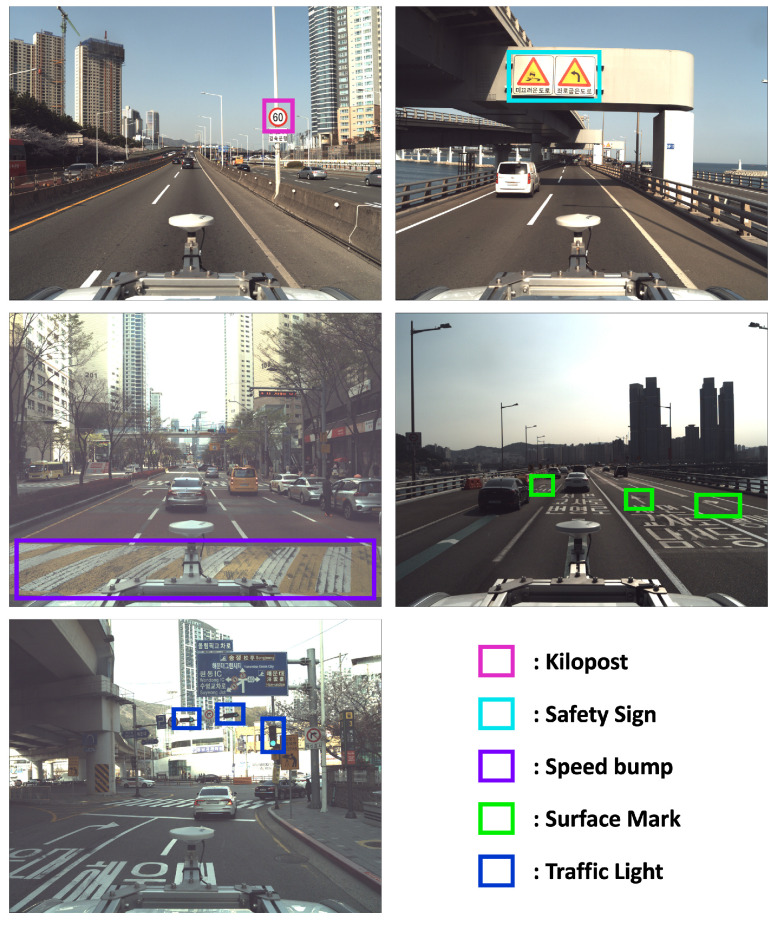
Selected objects from registered objects in HD map (kilo post: signs indicating speed limits, Safety sign: signs of alarming hazardous, Speed bump: bumps installed on the road to force a vehicle to slow down, surface mark: marks drew on the road, especially arrows, traffic light: devices that provide traffic signals to vehicles or people).

**Figure 3 sensors-24-05191-f003:**
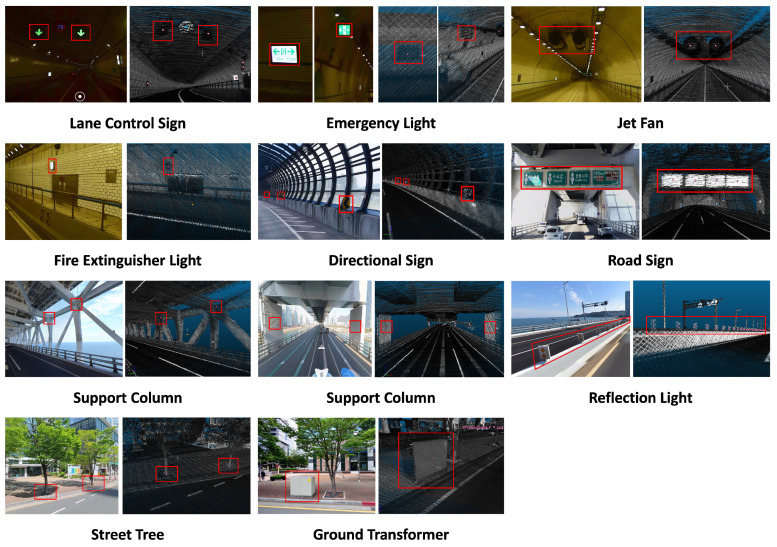
Additional objects.

**Figure 4 sensors-24-05191-f004:**
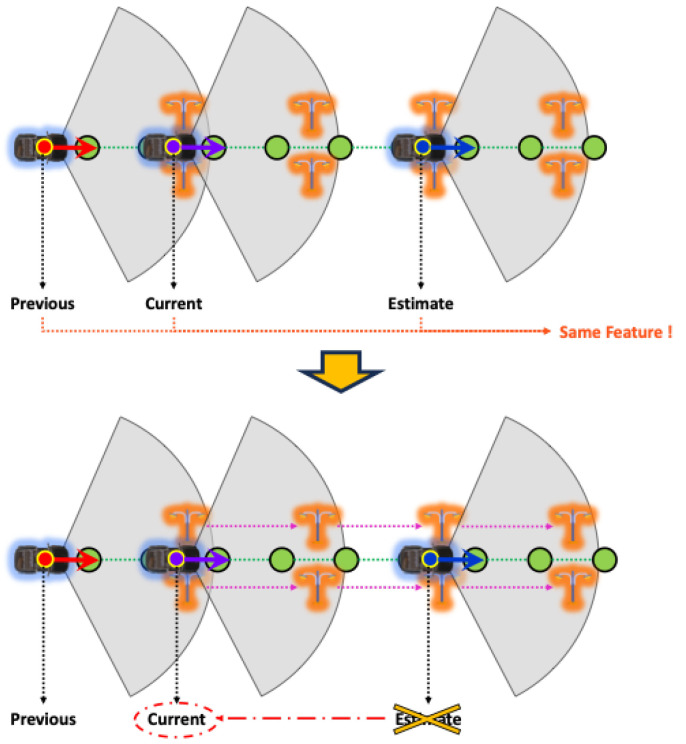
The problem of repeated objects. The same objects (orange objects) are repeated a long way on the links (green circles). In this case, the estimation from the previous to the current vehicle position (upper) can be wrong. If we know the relationship between objects, the estimation can be modified because we know that we estimated the wrong place (below).

**Figure 5 sensors-24-05191-f005:**
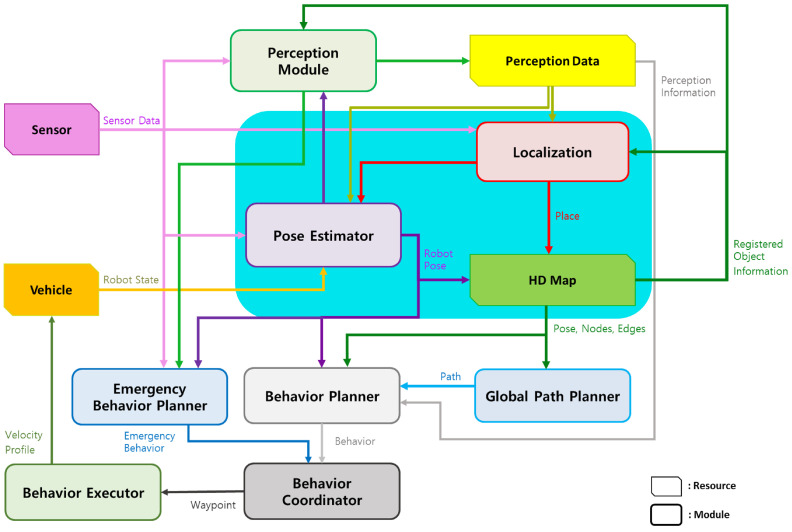
Autonomous driving system.

**Figure 6 sensors-24-05191-f006:**
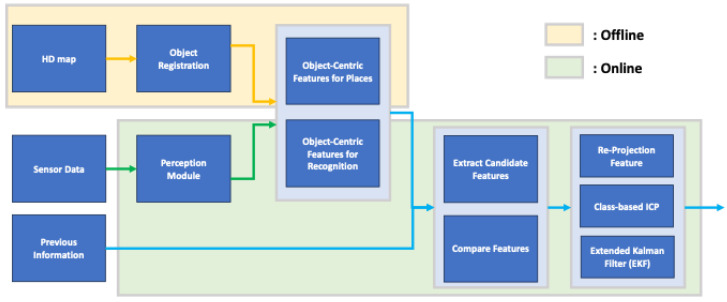
The total structure of our proposed method.

**Figure 7 sensors-24-05191-f007:**
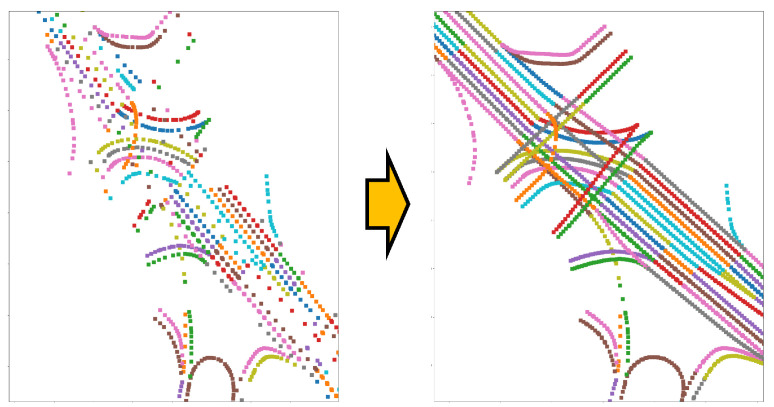
Link interpolation example. (**Left**) Basic link points registered in the HD map. (**Right**) Link points densely filled after link interpolation.

**Figure 8 sensors-24-05191-f008:**
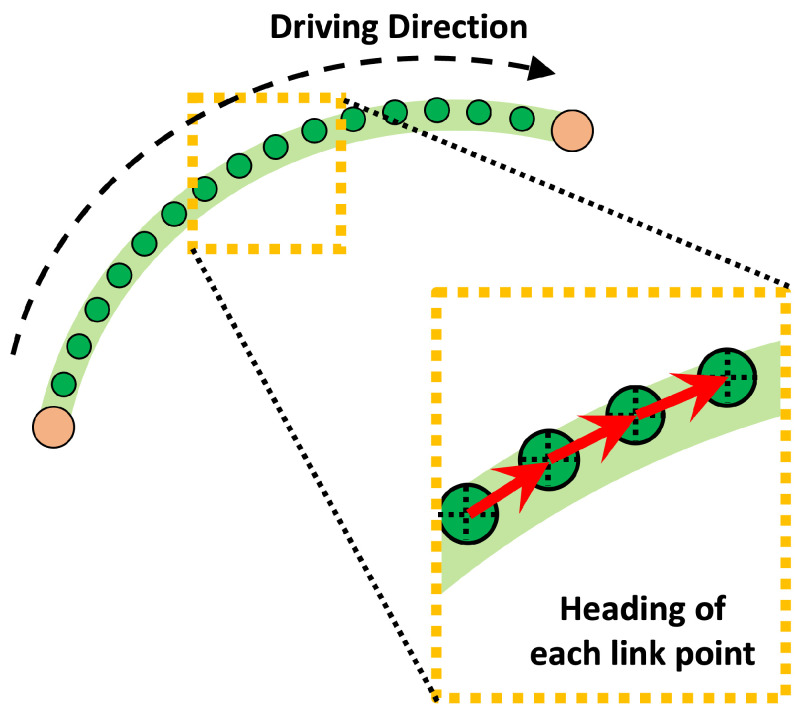
Vehicle heading examples. The heading (red arrow) of each link is decided by its following directional link (green circle).

**Figure 9 sensors-24-05191-f009:**
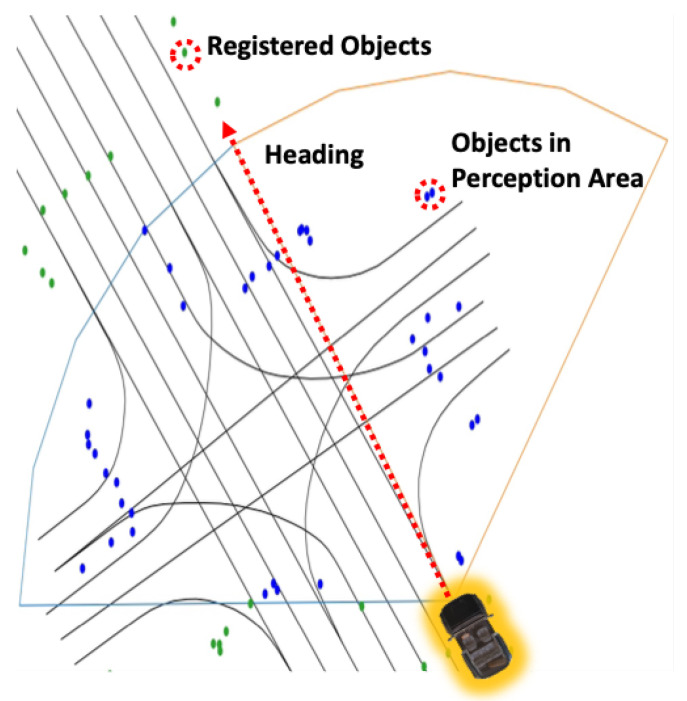
Perception coverage example based on the heading. All the lines indicate roads, and all the circles are registered objects. The blue circles presented are the registered objects included in the perception coverage.

**Figure 10 sensors-24-05191-f010:**
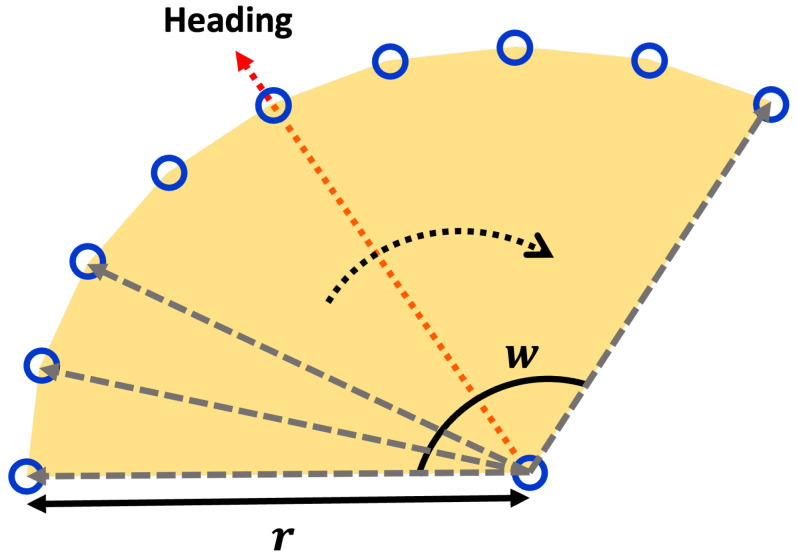
Making perception coverage based on the heading and the perception radius. The blue circles are composed to make the perception coverage points based on the perception angle *w* and the perception radius *r* from the left side to the right side. The perception coverage points have a specific interval based on the division number of *w*.

**Figure 11 sensors-24-05191-f011:**
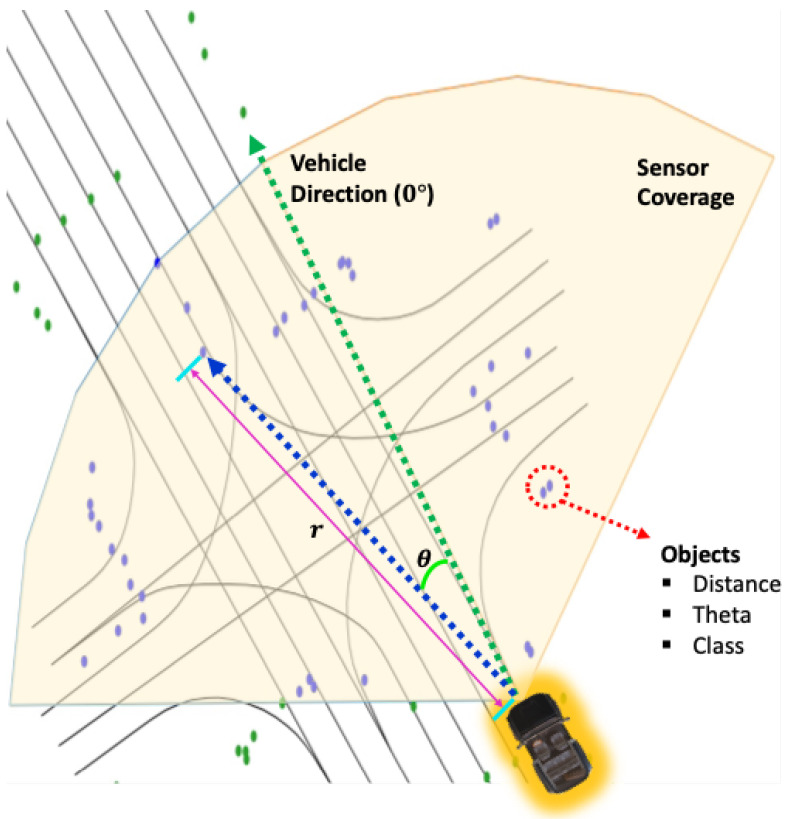
Object information is in the perception coverage. The vehicle’s heading is set to 0° for the polar coordinates of objects. Each object has distance, theta, and class information.

**Figure 12 sensors-24-05191-f012:**
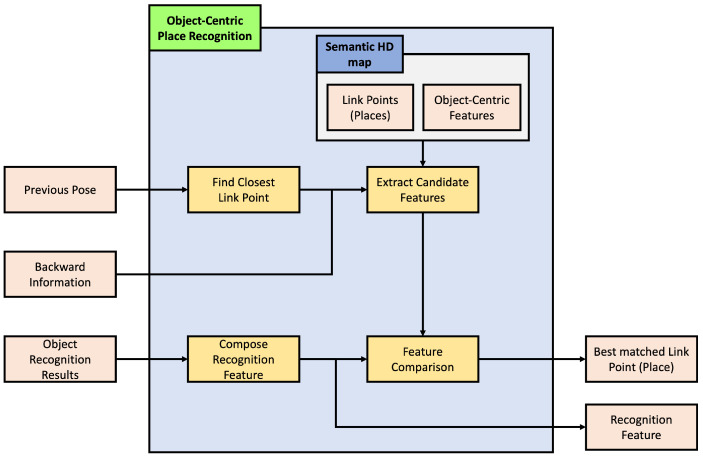
Object-centric place recognition process.

**Figure 13 sensors-24-05191-f013:**
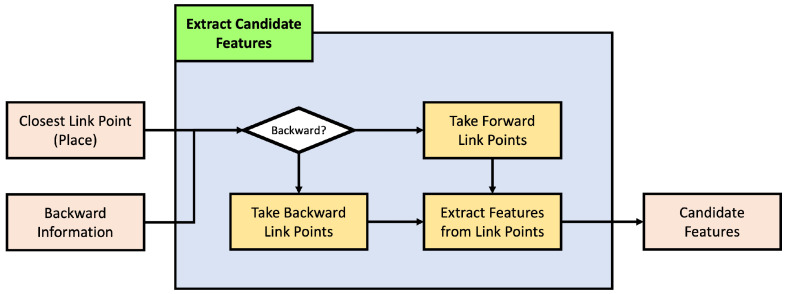
Extract candidate features process.

**Figure 14 sensors-24-05191-f014:**
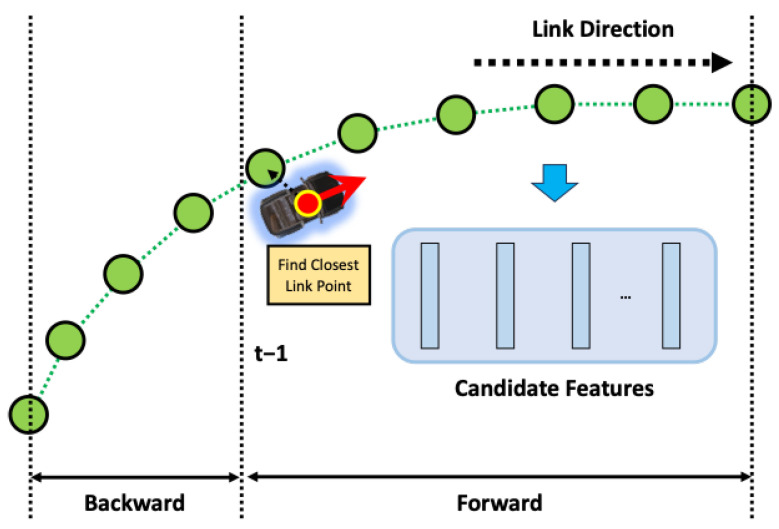
Sample of candidate feature extraction. Based on the previous position of the vehicle, find the closest link point and take candidate features based on the driving direction.

**Figure 15 sensors-24-05191-f015:**
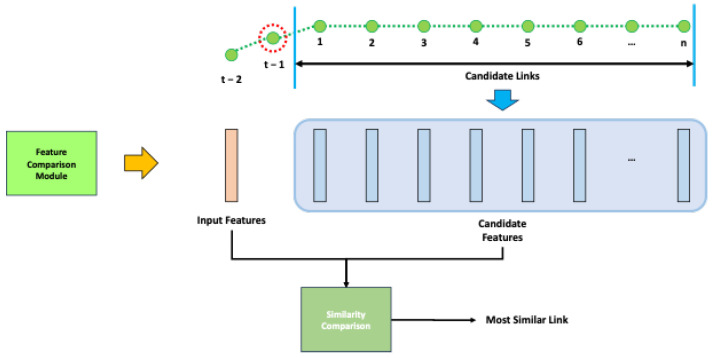
Feature comparison example: The input feature from the perception information is compared with candidate features to find the link with the most similar feature.

**Figure 16 sensors-24-05191-f016:**
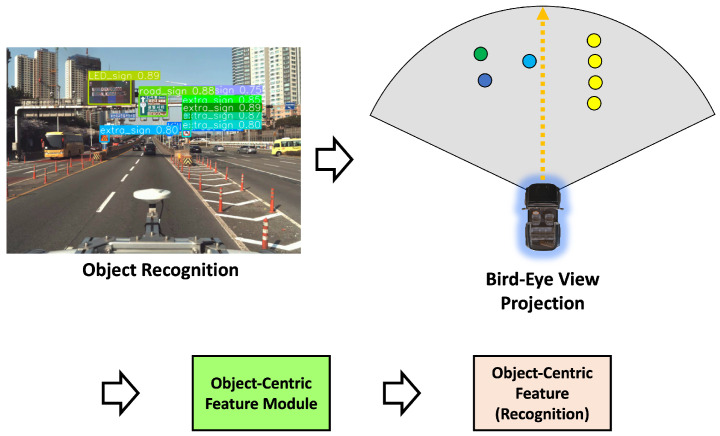
Generate recognition features. The object recognition results are projected as a bird-eye view. Then, the recognition feature is composed through the object-centric feature module.

**Figure 17 sensors-24-05191-f017:**
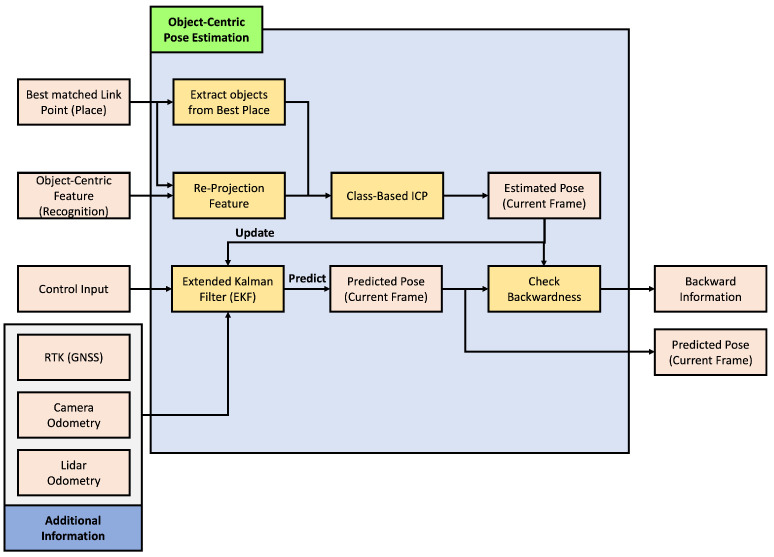
Object-centric pose estimation process.

**Figure 18 sensors-24-05191-f018:**
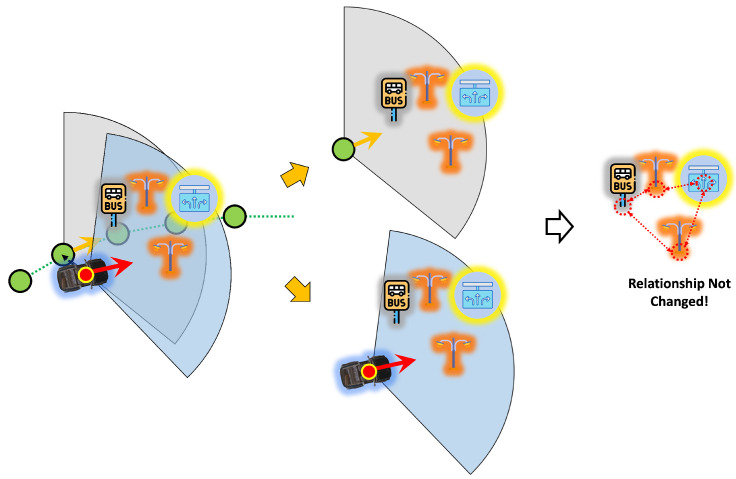
Difference between link point features and recognition features. The gray area is the perception coverage based on the link, and the blue area is the perception coverage based on the actual vehicle position. Ideally, the relationships between objects are the same in both areas.

**Figure 19 sensors-24-05191-f019:**
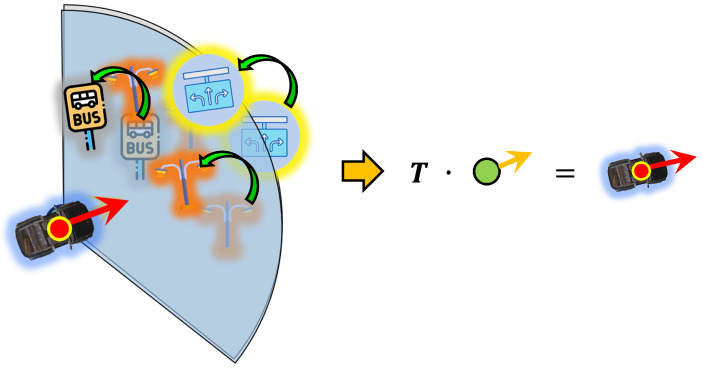
Class-based ICP example. Even though the relationships between objects are the same, there is a difference when the gray and blue areas are overlapped as on the left. This difference *T* is how the actual vehicle is transformed from the link.

**Figure 20 sensors-24-05191-f020:**
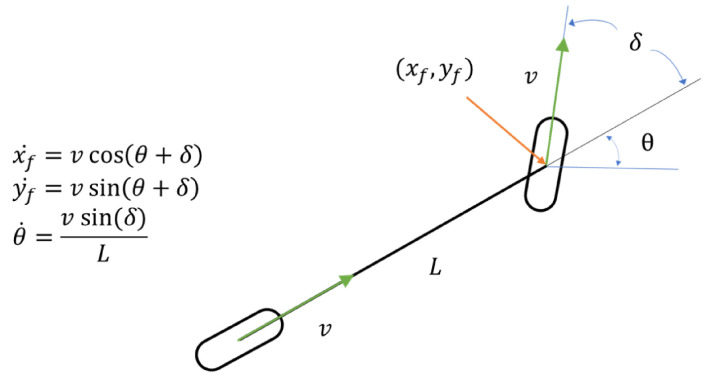
Vehicle kinematics.

**Figure 21 sensors-24-05191-f021:**
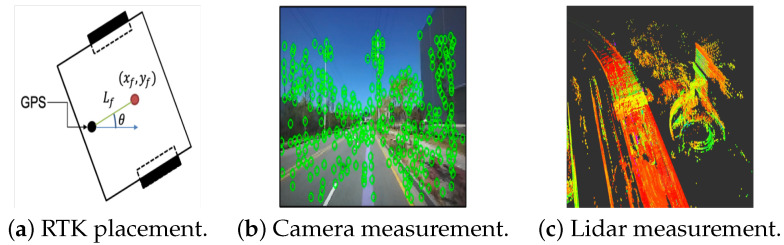
Measurements for the additional information. The RTK sensor is in place apart from the center of the vehicle, and we need to correct the RTK information as the center of the vehicle (**a**). The camera and Lidar measurements are composed of the difference between the previous and current features (**b**,**c**) of each sensor.

**Figure 22 sensors-24-05191-f022:**
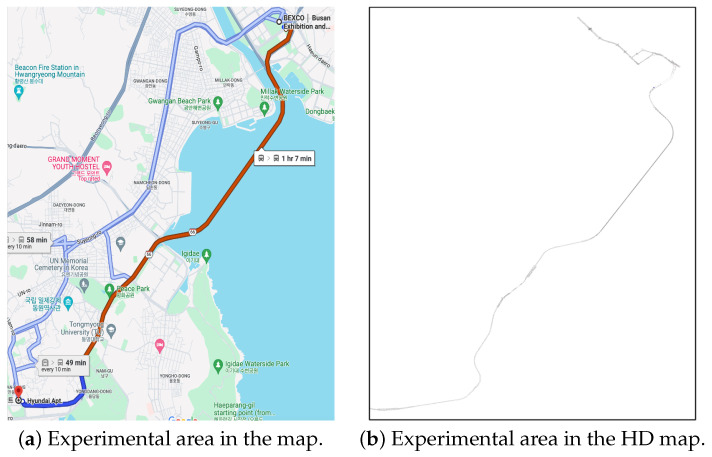
Experimental area.

**Figure 23 sensors-24-05191-f023:**
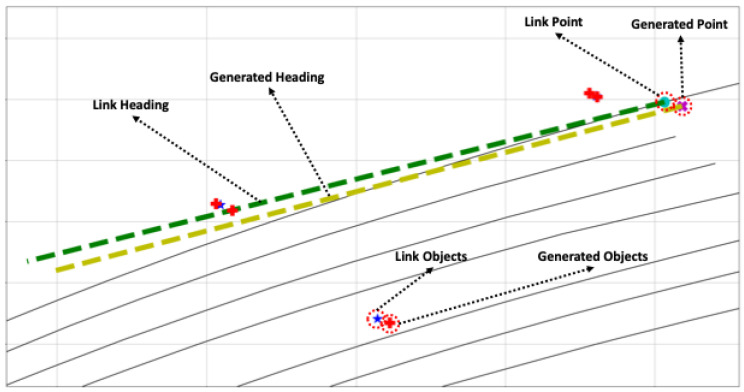
Imaginary object recognition example. Based on the link point (cyan circle), we generated a fake point (magenta cross mark). Also, based on the link heading (green dashed line), we generated a fake heading (yellow dashed line). Moreover, we generated fake objects (red Greek cross marks) and were concerned about the fake coverage from the fake heading and link objects (blue stars).

**Figure 24 sensors-24-05191-f024:**
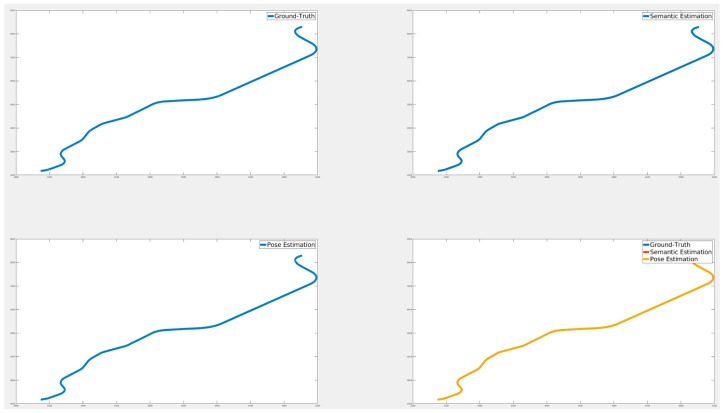
Ideal case result (without RTK). Ideally, the object-centric estimation (semantic estimation, upper right) and the EKF estimation (pose estimation, lower left) are nearly perfectly matched (lower right) with ground truth (upper left).

**Figure 25 sensors-24-05191-f025:**
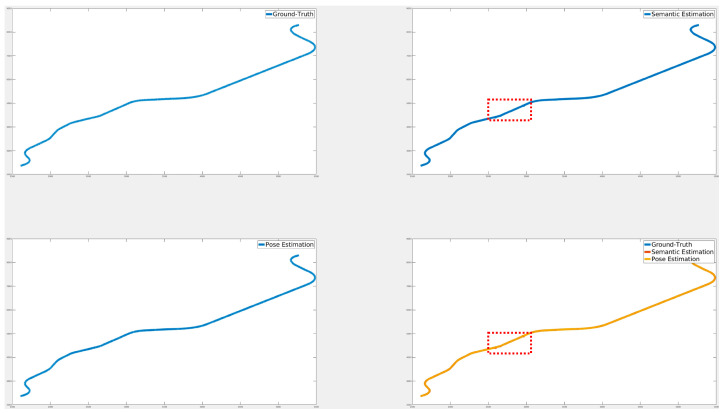
Slight case result (without RTK). The object-centric estimation (semantic estimation, upper right) is estimated unstably in a particular section (red dotted box). However, EKF estimation (pose estimation) was corrected finely (bottom left) and matched with ground truth (bottom right).

**Figure 26 sensors-24-05191-f026:**
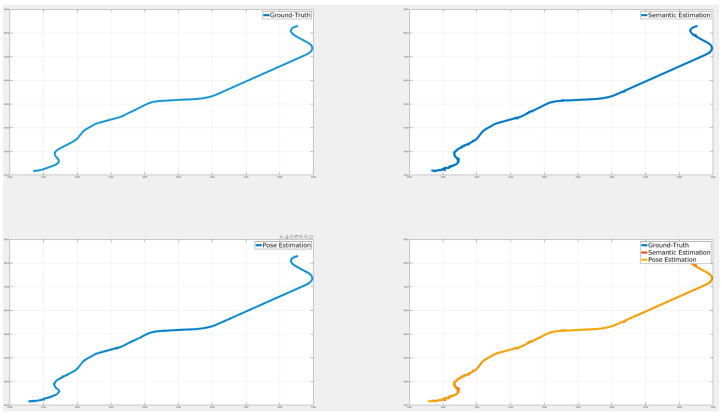
Pronounced case result (without RTK). As shown here, the object-centric estimation (semantic estimation) is vibrated because the object recognition results are unstable (upper right). Even the EKF estimation (pose estimation) tried to correct these vibrated estimations, but it couldn’t be corrected perfectly (bottom left) and estimated far from the ground truth (bottom right).

**Figure 27 sensors-24-05191-f027:**
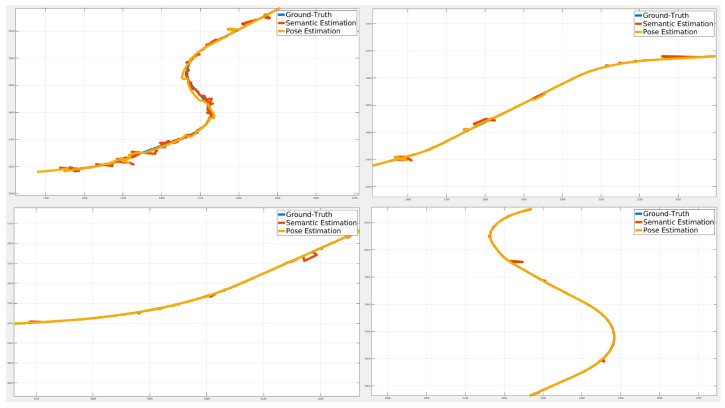
Pronounced case result (without RTK, detailed). These are zoomed sections from Figure 26. Here, the EKF estimation (pose estimation) is mostly performed to match the ground truth; even the object-centric estimation (semantic estimation) estimates its positions poorly.

**Figure 28 sensors-24-05191-f028:**
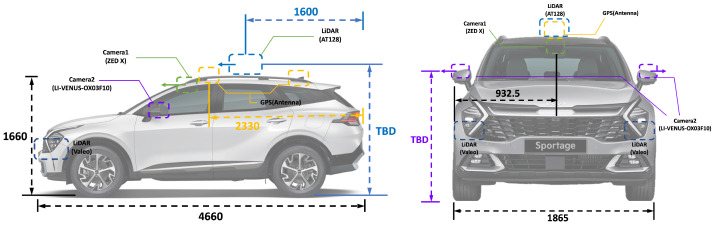
Sensor placement of the experimental vehicle.

**Figure 29 sensors-24-05191-f029:**
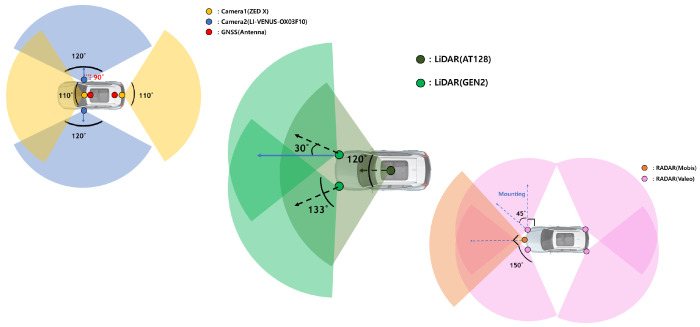
Sensor coverage of the experimental vehicle. Each colored coverage presents each sensor’s horizontal coverage. Depending on the sensors, they have different horizontal fields of view(FOV), as shown here.

**Figure 30 sensors-24-05191-f030:**
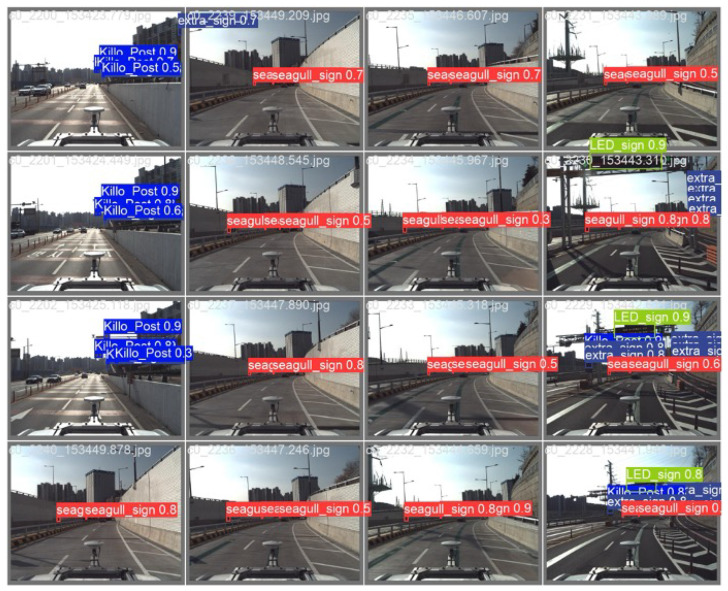
Object detection results.

**Figure 31 sensors-24-05191-f031:**
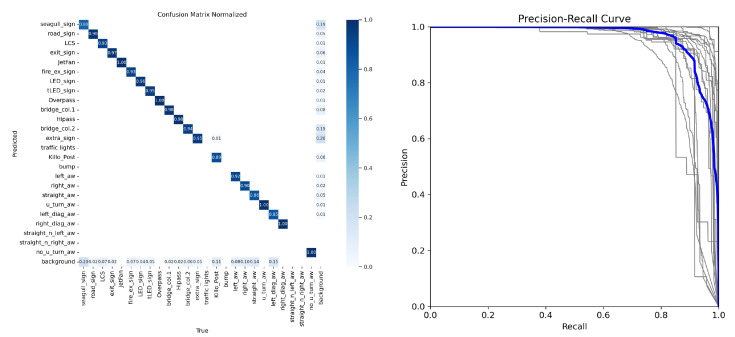
Object detection confusion metrics and P-R curve.

**Figure 32 sensors-24-05191-f032:**
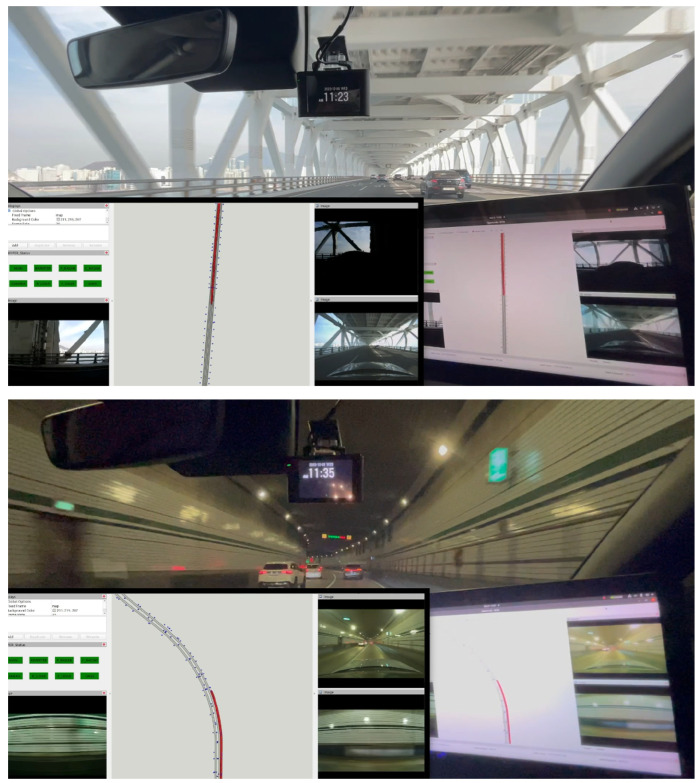
Actual experiments on the bridge (top) and tunnel (bottom).

**Figure 33 sensors-24-05191-f033:**
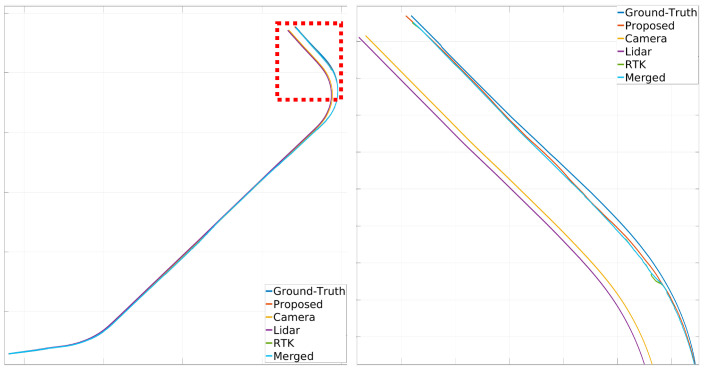
Actual experiment results (bridge). (Left) Half section includes bridge area (red box). (Right) Expanded bridge area from left.

**Figure 34 sensors-24-05191-f034:**
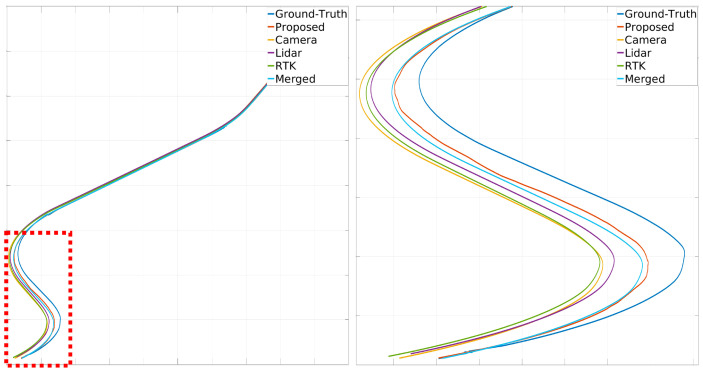
Actual experiment results (tunnel). (Left) Half section includes tunnel area (red box). (Right) Expanded tunnel area from left.

**Table 1 sensors-24-05191-t001:** Included information of nodes and links.

**Node**	ID	
	Point	
**Link**	ID	
	RoadType	1: General road
		2: Tunnel
		3: Bridge
		4: Underground road
		5: Elevated highway
	R_LinkID	
	L_LinkID	
	FronNodeID	
	ToNodeID	
	Points	

**Table 2 sensors-24-05191-t002:** Selected objects and their information (simplified).

**Safety Sign**	ID	
	Point	
**Traffic Light**	ID	
	Point	
**Kilopost**	ID	
	Point	
**Speed Bump**	ID	
	Point	
**Surface Mark**	ID	
		1: Arrow
	Type	2: Crosswalk
	Points	

**Table 3 sensors-24-05191-t003:** Accuracy criteria for additional objects.

Absolute Accuracy (95% Confidence Interval)		Maximum Error	
**Horizontal Position**	**Vertical Position**	**Horizontal Position**	**Vertical Position**
≤0.1 (m)	≤0.1 (m)	≤0.2 (m)	≤0.2 (m)

**Table 4 sensors-24-05191-t004:** TOSM data types for each object.

Datatype	Property	Domain	Range
Symbolic	ID	Object	String
	Type	Object	Int
	Group	Object	Int
	Count	Object	Int
Explicit	Coordinate	Object	(x, y)
	CoordinateFrame	Object	String
Implicit	isKeyObject	Object	Boolean
	isMovable	Object	Boolean
	isRepeated	Object	Boolean

**Table 5 sensors-24-05191-t005:** Relation types between objects.

Object Relation	Type	Domain	Range
Spatial Relation	fromObjectID	Object	Object
	toObjectID	Object	Object

**Table 6 sensors-24-05191-t006:** Types of additional objects.

**Additional Objects**	Type
	11: Directional Sign
	12: Road Sign
	21: Lane Control Sign
	22: Emergency Light
	23: Jet Fan
	24: Fire Extinguisher Light
	31: Support Column
	32: Reflection Light
	33: Street Tree
	34: Ground Transformer

**Table 7 sensors-24-05191-t007:** Number of all registered objects.

**Safety Sign**	900	
**Surface Mark**	**Arrow**	129
	**Crosswalk**	537
**Traffic Light**	262	
**Speed Bump**	36	
**Additional Objects**	**Directional Sign**	31
	**Road Sign**	433
	**Lane Control Sign**	502
	**Emergency Light**	121
	**Jet Fan**	273
	**Fire Extinguisher Light**	42
	**Support Column**	241
	**Reflection Light**	253
	**Street Tree**	965
	**Ground Transformer**	254

**Table 8 sensors-24-05191-t008:** Object-centric pose estimation results in simulation (with RTK).

Trial	Cases					
	Ideal		Slight		Pronounced	
	Long. (m)	Lat. (m)	Long. (m)	Lat. (m)	Long. (m)	Lat. (m)
1	0.0970	0.1692	0.2100	0.3586	0.5711	0.7133
2	0.1157	0.1995	0.3781	0.2137	0.5323	0.5263
3	0.1441	0.1477	0.3389	0.4151	0.5544	0.3488
4	0.1657	0.1025	0.3269	0.2513	0.4450	0.8318
5	0.1347	0.1407	0.3114	0.3404	0.7170	0.3340
6	0.1095	0.2051	0.1711	0.2845	0.6837	0.3125
7	0.1660	0.1190	0.1968	0.2920	0.5176	0.6936
8	0.1565	0.1321	0.3733	0.2399	0.6686	0.4062
9	0.1577	0.1205	0.2380	0.3988	0.6623	0.3523
10	0.1471	0.1313	0.3799	0.2701	0.7180	0.3374
Mean	0.1394	0.1468	0.2924	0.3064	0.6070	0.4856
Total	0.1431		0.2994		0.5463	

**Table 9 sensors-24-05191-t009:** Object-centric pose estimation results in simulation (without RTK).

Trial	Cases					
	Ideal		Slight		Pronounced	
	Long. (m)	Lat. (m)	Long. (m)	Lat. (m)	Long. (m)	Lat. (m)
1	0.4538	0.2450	0.4700	0.6480	2.6065	1.9879
2	0.2540	0.4264	0.8383	0.3771	9.7814	10.0362
3	0.4345	0.3217	0.6929	0.3363	5.7851	6.0601
4	0.3164	0.4584	0.3988	0.6350	4.1790	0.8166
5	0.2786	0.4496	0.3389	0.7549	7.6729	5.0775
6	0.4850	0.2724	0.8350	0.3524	4.5744	6.4828
7	0.4437	0.3017	0.6913	0.3919	9.1897	10.1281
8	0.4423	0.3323	0.7755	0.2921	2.6119	6.9593
9	0.2904	0.4604	0.4171	0.7713	7.6462	8.0244
10	0.3137	0.3771	0.5049	0.5185	5.1110	5.5926
Mean	0.3712	0.3645	0.5963	0.5078	5.9158	6.1166
Total	0.3679		0.5520		6.0162	

**Table 10 sensors-24-05191-t010:** Compare pose estimation results. The best results are expressed in bold. As shown here, the proposed method outperforms other results.

Area	Methods									
	RTK		Camera		LiDAR		Camera+LiDAR		Proposed	
	Long. (m)	Lat. (m)	Long. (m)	Lat. (m)	Long. (m)	Lat. (m)	Long. (m)	Lat. (m)	Long. (m)	Lat. (m)
Bridge	1.7640	1.9497	3.9769	7.9231	5.4016	10.3363	0.7714	0.9373	**0.6507**	**0.7422**
Tunnel	1.9940	6.5235	1.9135	7.0682	1.9940	6.4776	1.0480	3.2426	**0.9299**	**2.6461**
Mean	1.8790	4.2366	2.9452	7.4957	3.6978	8.4070	0.9097	2.0901	**0.7903**	**1.6942**
Total	3.0578		5.2204		6.0524		1.4998		**1.2422**	

## Data Availability

No new data were created or analyzed in this study. Data sharing is not applicable to this article.

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
