# Peer review of "An Object-Centric Hierarchical Pose Estimation Method Using Semantic High-Definition Maps for General Autonomous Driving"

_sensors, 2024, doi:10.3390/s24165191_

Round 1

Reviewer 1 Report

Comments and Suggestions for Authors

A method to overcome these challenges by leveraging objects registered in a high-precision map is proposed in this paper. This paper is well-organized. I have the following comments, which may be helpful to the authors to improve the quality of the paper.

1. In reference 19, the journal should be corrected in a right way.

2. In reference 24 and 58, the pages of reference should be presented.

3. In page 21, an extended Kalman filter for pose estimation is incorporated to the module, what is main difficult meet during this time?

4.In page 23, the corresponding letter should be explained, for example  p_{t^{*}_{j}}should be explained.

5. It should add some comparisons with the existing related papers to illustrate the innovation of this article.

6.In page 35, line834, the reference should be presented.

7.Yolo v10 can be applied, why authors adopt Yolov8 to train.

8.More experimental data should be presented to explain the effectiveness of the method.

9.feature fusion for object detection at one map, it may helpful to improve the result of this paper.

Comments on the Quality of English Language

1. In reference 19, the journal should be corrected in a right way.

2. In reference 24 and 58, the pages of reference should be presented.

3.In page 23, the corresponding letter should be explained, for example  p_{t^{*}_{j}}should be explained.

4.In page 35, line834, the reference should be presented.

Author Response

Comments 1:
In reference 19, the journal should be corrected in a right way.

Response: 
The name of the reference Journal, Reference 19, has been corrected. In addition, other reference journal names have been checked and corrected.

Comments 2: 
In reference 24 and 58, the pages of reference should be presented.

Response: 
As a result of verification,
- Reference 24 is 'J. Imaging 2022, 8(4), 106; https://doi.org/10.3390/jimaging8040106'; the page number is 106, so no separate correction is necessary.
- Reference 58 has been rewritten by adding the page number to fit the format.

Comments 3:
In page 21, an extended Kalman filterfor pose estimation is incorporated to the module, what is main difficult meet during this time?

Response:
Although we configured the modified EKF in this way when applied, the input cycles of all sensors and the cycles of the estimation results are different, and the cycles can also change instantaneously depending on the system's stability. If we wait for all data to be input, not only will the measurement update be slow, but errors may occur during the update because the times of each data are different. Therefore, we used the measurement vector configured above as it is but configured the update part to operate whenever each sensor and estimation result data is acquired. (27 page, 713 ~ 719)

Comments 4:
In page 23, the corresponding letter should be explained, for example  p_{t^{*}_{j}}should be explained.

Response:
I have added character descriptions for equations 1 ~ 3 on page 15 (438 ~. 444), 7 ~ 8 on page 17 (475 ~ 477), 9 ~ 13 on page 18 (505 ~ 510), and 18 ~ 22 on page 22 (571 ~ 589).

Comments 5:
It should add some comparisons with the existing related papers to illustrate the innovation of this article.

Response:
As one way to compare existing papers' methods, Table 10. is shown in this paper. This table shows a comparison table that includes the existing RTK-based driving method, camera feature, and lidar feature-based driving method. It compares the errors when our method was used alone or in combination, including the ORB method, the most famous camera feature methods, and the Fast-LIO method, which has fast and accurate performance among lidar feature methods. We also planned to compare it with other papers. However, we could not do so because the vehicle we designed and built is too different from ours, the reference code does not exist, or it is not easy to experiment with the reference code to fit our vehicle and environment. We plan to obtain and compare related data through future experiments.

Comments 6:
In page 35, line834, the reference should be presented.

Response:
We modified the reference for Yolo v8 (35 page, 872).

Comments 7:
Yolo v10 can be applied, why authors adopt Yolov8 to train.

Response:
When we experimented, we judged that the reference code for Yolo v10 was unstable for application and use in actual vehicles, so we adopted and used Yolo v8, which is the most stable and commonly used.

Comments 8:
More experimental data should be presented to explain the effectiveness of the method.

Response:
Unfortunately, it is difficult to obtain more experimental data because the related project has been completed, and it is difficult to drive and experiment with the vehicle. We plan to borrow a vehicle and conduct additional experiments to obtain more experimental data when conducting related projects in the future.

Comments 9:
feature fusion for object detection at one map, it may helpful to improve the result of this paper.

Response:

As mentioned, this paper must include the feature fusion for re-localization. When an RTK signal can be acquired, feature comparison is performed based on that location. However, there is a problem with all features of the entire map being compared if the location is lost momentarily when the RTK signal is not coming in. We are also aware of this. We are conducting future research on generating feature nodes and regional feature graphs by area for re-localization. As soon as the research is completed, we will publish a separate paper in a follow-up paper.

Reviewer 2 Report

Comments and Suggestions for Authors

This paper proposes an object-centered hierarchical pose estimation method to reduce dependence on RTK sensors for autonomous vehicles and enable stable and accurate pose estimation in various environments. It is noteworthy that we are trying to overcome the signal interference problem of RTK sensors by utilizing object information registered in high-precision maps. The paper demonstrates the effectiveness of the proposed method through experiments in simulation and real environments.

But there are a few things that need to be fixed

1.

First, the experiment is limited to a specific environment or simulation.

To resolve this, more real-world testing is needed, including various road conditions, weather, and lighting conditions.

Additionally, it would be nice to add performance evaluation on various road types such as city roads, highways, and rural roads.

2.

And in order to recognize a location based on an object, the reliability and accuracy of object recognition are important.

The performance of object recognition algorithms must be evaluated under various conditions, and the impact of recognition errors on pose estimation must be analyzed.

Methods or supplementary measures to reduce uncertainty in object recognition should be discussed.

3.

In order to perform pose estimation based on object recognition, experiments are needed to evaluate the computational complexity and real-time processing capabilities of RISM.

Please add a review of the real-time processing potential of the proposed method.

4.

This comment is an additional opinion, so please describe the author's judgment in detail.

Various sensors (e.g. LiDAR, radar) can be used for object recognition, and their performance has not been clearly verified.

It is necessary to evaluate the performance of the proposed method when integrated with other sensors, and to discuss specifically the data fusion method for each sensor.

It would be good to suggest a method to improve pose estimation performance based on multi-sensor fusion.

5.

Therefore, it is not clear whether the proposed method compares with other state-of-the-art techniques.

Performance comparison studies with other existing pose estimation methods are needed. Through this, the advantages and disadvantages of the proposed method should be clarified and areas that can be improved should be identified.

Author Response

Comments 1:
First, the experiment is limited to a specific environment or simulation.
To resolve this, more real-world testing is needed, including various road conditions, weather, and lighting conditions.
Additionally, it would be nice to add performance evaluation on various road types such as city roads, highways, and rural roads.

Response:
We also agree. When researching autonomous driving, researchers need to define ODD for the autonomous driving system and verify the system accordingly. However, since this paper aims to propose a posture estimation method that can be applied more generally to autonomous driving research, we did not necessarily divide the various conditions, and although it is not described in detail in the paper, the time, weather, and road conditions were different during the experiment. As suggested, we plan to conduct additional experiments based on various conditions, and we have applied for road driving permits in several regions to test various road types. The project is currently in the final stage, and we are waiting for this process as we need to rent a separate vehicle again for additional performance evaluation.

Comments 2:
And in order to recognize a location based on an object, the reliability and accuracy of object recognition are important.
The performance of object recognition algorithms must be evaluated under various conditions, and the impact of recognition errors on pose estimation must be analyzed.
Methods or supplementary measures to reduce uncertainty in object recognition should be discussed.

Response:
We concur with the need for continuous data collection and labeling under various conditions and road types and the importance of enhancing object recognition accuracy through additional learning. While we currently use Yolo v8, we are committed to increasing reliability by upgrading to the improved Yolo v10 or integrating another object recognition network that suits the situation. As you rightly pointed out, the reliability of object recognition is influenced by the uncertainty of recognition, which varies with conditions and situations. Therefore, we are also developing a method to generate object recognition reliability that aligns with conditions and situations and adjusts the weights when deriving results.

Comments 3:
In order to perform pose estimation based on object recognition, experiments are needed to evaluate the computational complexity and real-time processing capabilities of RISM.
Please add a review of the real-time processing potential of the proposed method.

Response:
In this paper, we experimented with a modified vehicle using the Orin kit from Nvidia instead of a general computer. As proposed in this paper, the speed of generating features through object recognition is remarkably fast, and the comparison process is equally swift due to its simple structure. It showed a speed higher than the camera frame (30hz). It was operated at a 10hz cycle according to the lidar frame, but it can be used faster if necessary. The computer can be applied at a lower cost, leading to a real-time configuration at an affordable price.

Comments 4:
This comment is an additional opinion, so please describe the author's judgment in detail.
Various sensors (e.g. LiDAR, radar) can be used for object recognition, and their performance has not been clearly verified.
It is necessary to evaluate the performance of the proposed method when integrated with other sensors, and to discuss specifically the data fusion method for each sensor.
It would be good to suggest a method to improve pose estimation performance based on multi-sensor fusion.

Response:
In the proposed paper, the object's distance and class are judged based on the information of the object recognized in the image after the camera-lidar and camera-radar calibration is completed and the feature is composed. As mentioned in 3, the Nvidia Orin kit is currently being used, and it needs to be applied due to the application problem of the real-time operation method and resource problem among the object recognition methods based on point clouds. When recognizing with only the camera, errors in the distance and class information occur due to class errors in object recognition and bounding box or segmentation errors, which can cause problems operating the method we proposed. Of course, if the reliability is low, this information is not used, but even if the reliability is high, misrecognition can always occur. Suppose the lidar object recognition and radar object estimation information are combined. In that case, a more precise distance is expected to be obtained, and class information can be supplemented to prevent such misrecognition. However, more resources can be consumed in this case, so we plan to research to divide the operation according to the applicable hardware. 

Comments 5:
Therefore, it is not clear whether the proposed method compares with other state-of-the-art techniques.
Performance comparison studies with other existing pose estimation methods are needed. Through this, the advantages and disadvantages of the proposed method should be clarified and areas that can be improved should be identified.

Response:
One of the ways to compare with the methods of existing papers is shown in Table 10. This table shows a comparison table that includes the existing RTK-based driving method and the camera feature and lidar feature-based driving method to which our proposed method is applied. We compared the errors when the ORB method, the most famous camera feature method, and the Fast-LIO method, which has fast and accurate performance among lidar feature methods, were used alone or in combination and when our method was applied. We also planned to compare it with other papers. However, we could not do it because the vehicle configuration we designed and built was too different from ours, the reference code did not exist, or it was not easy to experiment with the reference code to fit our vehicle and environment. We plan to acquire and compare related data through future experiments. In addition, we plan to improve recognition and attitude estimation performance through the method of 4. mentioned above.

Reviewer 3 Report

Comments and Suggestions for Authors

The manuscript titled, “An Object-Centric Hierarchical Pose Estimation Method Using Semantic HD Maps for General Autonomous Driving,” proposes a pose estimation method by utilizing HD maps to support autonomous driving. However, there are several shortcomings in the present form of the manuscript that needs a major revision. A few of the suggestions to improve the quality of the manuscript are as follows:

-        -  It is good practice to introduce the full form of abbreviations in abstract and in their first appearance of the manuscript. Authors are advised to do so in the revised manuscript for avoiding any difficulty in understanding.

-      - It seems that authors are confused in ADAS and ADS in context of autonomous vehicles. The same was reflected in line#40 and line#42 respectively. “… Levels 0 to 3 involve ADAS-level autonomous driving, … Levels 4 to 6 represent fully autonomous driving, where the driver does not need to be seated in the driver’s seat, and control is not required.” It is surprising to see level 6 in AV. Please refer these statements along with the SAE and NHTSA definitions and correct as applicable. Also revise the manuscript accordingly.

-          -   Many figures are poorly legible and their captions are inappropriate. In addition, Figure 1, it is recommended to indicate the links and nodes in figure for better comprehension of the future readers. Figure 2 has very thick bounding boxes which obstruct the other objects present in the image. It is recommended to reduce their thickness. Figure 7 is unable to understand because there is no legends and clarifications. Figure 19 is unable to visualise anything. Figure 22 has un-numbered subfigures. Figure 23 represents some map. Map should be in English for better comprehension etc. Please revise them with high-resolution clear images and change the captions to represent the purpose of the figure in the manuscript.

-    - Caption of table is usually placed on the top of the table. It is recommended to follow the format of the journal. Also, revise the captions to represent the purpose of the table.

-        -   It is common practice to not include citations in conclusion. If necessary authors can discuss them in discussion section separately.

-         - In line#289, authors have mentioned, “Additionally, we add the following properties to resolve … and isRepeated”. It is recommended to discuss how is this a problem to autonomous driving.

-        -   Further, in line#290, authors have justified, “Because the … in Figure 4.” It may be repeated however, sometimes required to estimate the location of the object when needed. Authors are recommended to elaborate on the issues of such repetitions.

-      - Reference section has very less recent references from year 2023, 2024. Author can add following relevant references to improve the manuscript:

 https://doi.org/10.1016/bs.adcom.2023.04.006    https://doi.org/10.3390/s21237914 Comments on the Quality of English Language

Authors should thoroughly proofread the manuscript to avoid the grammatical and typographic mistakes.

Author Response

Comments 1:
It is good practice to introduce the full form of abbreviations in abstract and in their first appearance of the manuscript. Authors are advised to do so in the revised manuscript for avoiding any difficulty in understanding.

Response:
As mentioned, we revised our abstract to improve understanding. Thanks for the comment (lines 5, 10).

Comments 2:
It seems that authors are confused in ADAS and ADS in context of autonomous vehicles. The same was reflected in line#40 and line#42 respectively. “… Levels 0 to 3 involve ADAS-level autonomous driving, … Levels 4 to 6 represent fully autonomous driving, where the driver does not need to be seated in the driver’s seat, and control is not required.” It is surprising to see level 6 in AV. Please refer these statements along with the SAE and NHTSA definitions and correct as applicable. Also revise the manuscript accordingly.

Response:
As mentioned, we revised the phrase correctly. Thanks for the comment (lines 40 ~ 49).

Comments 3:
Many figures are poorly legible and their captions are inappropriate. In addition, Figure 1, it is recommended to indicate the links and nodes in figure for better comprehension of the future readers. Figure 2 has very thick bounding boxes which obstruct the other objects present in the image. It is recommended to reduce their thickness. Figure 7 is unable to understand because there is no legends and clarifications. Figure 19 is unable to visualise anything. Figure 22 has un-numbered subfigures. Figure 23 represents some map. Map should be in English for better comprehension etc. Please revise them with high-resolution clear images and change the captions to represent the purpose of the figure in the manuscript.

Response:
We changed and added a caption description for Figure 1, reduced object line thickness for Figure 2, added a caption description for Figure 7, deleted Figure 19 as unnecessary, revised the sub-figure for Figure 21, and revised Figure 22 to an English map.

Comments 4:
Caption of table is usually placed on the top of the table. It is recommended to follow the format of the journal. Also, revise the captions to represent the purpose of the table.

Response:
Revised caption placement of tables to the upper left and changed table caption for tables 1, 2, 4, 5, 7, 8, 9.

Comments 5:
It is common practice to not include citations in conclusion. If necessary authors can discuss them in discussion section separately.

Response:
We excluded citations from the conclusion and revised the sentence accordingly.

Comments 6:
In line#289, authors have mentioned, “Additionally, we add the following properties to resolve … and isRepeated”. It is recommended to discuss how is this a problem to autonomous driving.

Response:
We revised the line from 308 to 322.

Comments 7:
Further, in line#290, authors have justified, “Because the … in Figure 4.” It may be repeated however, sometimes required to estimate the location of the object when needed. Authors are recommended to elaborate on the issues of such repetitions.

Response:
We revised the line from 308 to 322 for a detail description.

Comments 8:
Reference section has very less recent references from year 2023, 2024. Author can add following relevant references to improve the manuscript: https://doi.org/10.1016/bs.adcom.2023.04.006, https://doi.org/10.3390/s21237914

Response:
We included the suggested reference material and added reference materials for 2023~2024 to the last paragraph of 'Related Works' (lines 185 ~ 202).

Round 2

Reviewer 1 Report

Comments and Suggestions for Authors

This  paper can be accepted.

Comments on the Quality of English Language

This  paper can be accepted.

Author Response

Comments 1:
This paper can be accepted.

Response:
Thanks for your reviews and all the comments on our article.

Reviewer 2 Report

Comments and Suggestions for Authors

The responses are appropriate and provide a detailed, thoughtful acknowledgment of the reviewer's comments. They outline both current approaches and future plans, demonstrating a commitment to thorough research and continuous improvement. The responses are clear, specific, and well-justified, making them suitable and effective in addressing the reviewer's concerns.

Comments on the Quality of English Language

Minor editing of English language required

Author Response

Comments 1:
The responses are appropriate and provide a detailed, thoughtful acknowledgment of the reviewer's comments. They outline both current approaches and future plans, demonstrating a commitment to thorough research and continuous improvement. The responses are clear, specific, and well-justified, making them suitable and effective in addressing the reviewer's concerns.

Response:
Thanks for your review and all the comments on our manuscript. 

Comments 2:
Minor editing of English language required

Response:
We corrected English more accurately in our manuscript.

Reviewer 3 Report

Comments and Suggestions for Authors

Most of the comments have been addressed. However, a few corrections are still remaining. Such as:

- This reference is still missing in the revised version (https://doi.org/10.3390/s21237914).

- There are some typos and grammatical errors remaining (e.g. page#11; line#336 "... Chapter 3, ...". What is Chapter 3? and where is it?). It is recommended to thoroughly proofread and correct the manuscript.

Comments on the Quality of English Language

There are some typos and grammatical errors remaining (e.g. page#11; line#336 "... Chapter 3, ...". What is Chapter 3? and where is it?). It is recommended to thoroughly proofread to eliminate the language related errors present in the manuscript.

Author Response

Comments 1:
 This reference is still missing in the revised version (https://doi.org/10.3390/s21237914).

Response:
We added the suggested reference (line 202 ~ 204).

Comments 2:
There are some typos and grammatical errors remaining (e.g. page#11; line#336 "... Chapter 3, ...". What is Chapter 3? and where is it?). It is recommended to thoroughly proofread and correct the manuscript.

Response:
We modified line 338 properly. And also we corrected English in the manuscript.